

# Technical Note: Improved volume and derived value calculations for polished zircon

Barra A. Peak[1,2]

[1]Department of Geological Sciences, University of Colorado Boulder, Boulder, CO 80309, USA
[2] Now at Department of Earth and Planetary Sciences, University of Texas at Austin, Austin, TX 78712, USA

*Correspondence to*: Barra A. Peak (barra.peak@jsg.utexas.edu)

**Abstract.** Polishing mounted mineral crystals prior to bulk grain (U-Th)/He thermochronology analysis offers many advantages for characterizing and subsampling each grain via in situ methods to obtain the maximum geologically relevant information. However, polishing introduces complications for calculating grain volume, on which many derived (U-Th)/He data partially depend. Impacted data include isotope concentrations, effective uranium (a proxy for radiation damage), and alpha-ejection correction factors ($F_T$) which are used to correct (U-Th)/He dates. These derived data are integral to interpreting (U-Th)/He dates; without a way to accurately calculate these values for polished grains, the benefits of polishing and in situ measurements can be greatly reduced or negated. This reality has resulted in many studies forgoing polishing and thus missing potentially important data. To address this issue, this paper presents a set of equations encoded in an R script to calculate volume and derived values for polished zircon that can be easily integrated into existing workflows for bulk grain (U-Th)/He analysis.

## 1 Introduction

(U-Th)/He thermochronology dates and associated derived data are in part a function of mineral grain volume (V) (Cooperdock et al., 2019; Flowers et al., 2022; Zeigler et al., 2023, 2024). These derived data, including alpha-ejection correction factors ($F_T$), effective uranium (eU), and parent isotope concentrations, are essential for interpreting dates and making other geological inferences. $F_T$ values applied to account for He lost through alpha-ejection directly affect the reported (U-Th)/He dates (Farley, 2002), eU affects how the thermal history of the grain is interpreted (e.g., Guenthner et al., 2013), and other data such as isotope concentrations may be used to characterize additional aspects of the samples' geologic history (e.g., sediment recycling history, Dröllner et al., 2022). Accurate V calculation is therefore critically important. However, many applications of (U-Th)/He thermochronology, such as detrital zircon applications, benefit from or require mounting and polishing crystals for in situ analyses to characterize chemical zonation, rare-earth element abundances, U-Pb or other geochronology data, etc. Polishing removes part of the grain making calculation of V using standard methods more complicated or impossible.



The impact of polishing on $F_T$ has received the most attention and previous attempts at corrections (e.g., Marsden et al., 2021; Reiners, 2007; Reiners et al., 2005). A common approach to simplify $F_T$ corrections is to polish grains to a plane of symmetry (e.g., halfway), such that the $F_T$ value of the fragment is the same as the $F_T$ value for the entire grain (e.g., Reiners, 2007). However, polishing exactly halfway is often extremely difficult, if not impossible, and inaccuracy in polishing depth

can result in $F_T$ uncertainty greater than $1\sigma = 5$ % (Marsden et al., 2021). Existing corrections to $F_T$ that do not rely on polishing halfway (Reiners, 2007; Reiners et al., 2005) do not provide corrections for other values derived from V. Corrections to these values are arguably of as great or greater significance than $F_T$ since they impact all (U-Th)/He analyses, not just those subject to $F_T$ corrections (interior grain fragments and some detrital grains are not subject to alpha-ejection) and radiation damage (eU) is now known to heavily impact how (U-Th)/He dates should be interpreted (e.g., Guenthner et

al., 2013). To address the lack of holistic approach to volume-derived data for polished zircon, I present a set of equations coded in an R script (see Code Availability). Values calculated include V, surface area (SA), volume-to-surface area-equivalent spherical radius ($R_{SV}$), mass (M), parent isotope concentrations, eU, $F_T$, and $F_T$-equivalent spherical radius ($R_{FT}$). Values are generally independent of the degree of polishing, although an estimate of the material removed is required in some cases. In general, this method involves fewer assumptions and potential sources of uncertainty than other methods

currently employed to assign $F_T$ values and is the only explicit consideration of other volume-derived values for polished zircon that I am aware of.

## 2 Required measurements and grain classification

Methods for calculating V and SA involve relating two-dimensional (2D) grain measurements to idealized whole-grain geometries (e.g., Ketcham et al., 2011; Reiners et al., 2005). The calculations presented here for polished grains are similar

to calculations for whole grains but differ in several important ways. Conventional V and SA calculations require first classifying each grain according to its closest ideal geometry (e.g., Ketcham et al, 2011). This classification (ellipsoidal, tetragonal, or more rarely, cylindrical, in the case of zircon) depends on the original grain morphology and degree of abrasion or fragmentation. Interior fragments may be best described by geometries completely unrelated to the original grain. 2D measurements of the length (L) and width (W) of the grain are related to the relevant parameters associated with the

geometry classification: semi-axes or axes a, b, and c in the case of ellipsoids and tetragons, or radii and height r and h in the case of cylinders (Ketcham et al., 2011; Fig. 1a). Grain measurements are made using a binocular microscope with digital camera and microscope imaging software after removal from the mounting medium. Two sets of measurements ($L_1$, $W_1$ and $L_2$, $W_2$) are made orthogonal to each other by rotating the grain 90 ° (e.g., Peak et al., 2023).

For polished grains, 2D measurements are related to geometric parameters for the idealized geometry as a function of grain orientation to the polishing surface (perpendicular or parallel to c-axis) and polishing depth (< or > halfway) (Fig. 1b, c, Table 1). Equations for V and SA as a function of geometry, polishing orientation, and depth, if applicable, are modified





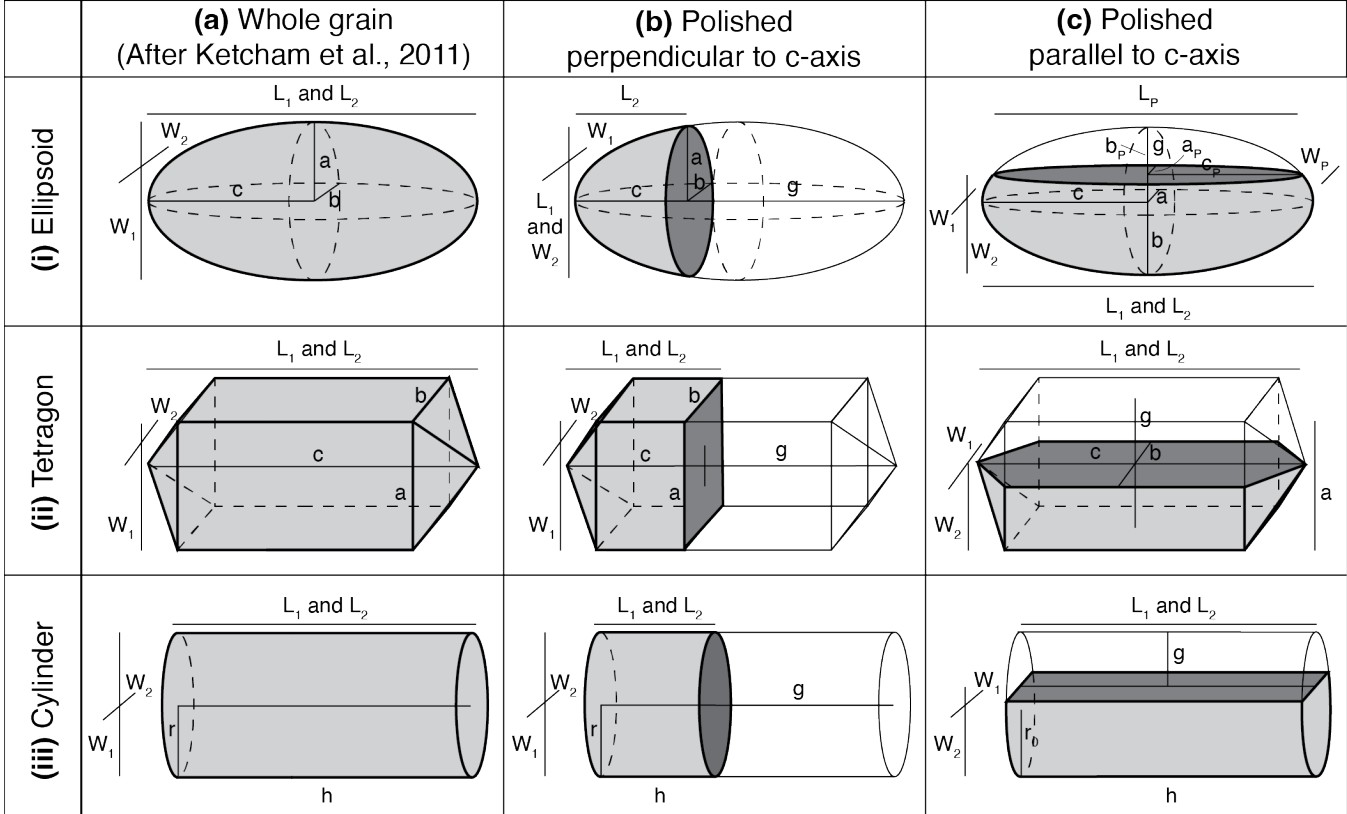

**Figure 01: Relationship between 2D grain measurements $L_1$, $L_2$, $W_1$, $W_2$, and semi-axes or axes a, b, c or r and h. (a) Whole grains (after Ketcham, et al., 2011). (b) grain fragments arising from polishing grains perpendicular to the c-axis. (c) grain fragments arising from polishing grains parallel to the c-axis. (i) ellipsoid idealized geometry. (ii) tetragon idealized geometry. (iii) cylinder idealized geometry. In all panels, light gray shaded region corresponds to the volume measured. Dark gray surface is polished surface not included in surface area.**

from equations for whole-grain geometries in Ketcham et al. (2011). Only external surfaces are subject to alpha-ejection, and thus the polished surface is not considered as part of SA in the calculation of $R_{SV}$ and $F_T$ (Fig. 1). For grains polished perpendicular to the c-axis (Fig. 1b) V and SA are relatively straightforward to calculate from measurements of the remaining grain: the degree of polishing, whether greater than, less than, or exactly halfway is irrelevant. For grains polished parallel to the c-axis (Fig. 1c) and less than halfway, V and SA calculations use polishing depth and measurements of the remaining grain to estimate the original grain V and SA and amount removed through polishing. Measurement of polishing depth is accomplished by mounting spherical glass beads alongside the grains and measuring the radius of the polished bead surface ($r_{BP}$) relative to the full bead radius ($r_B$) to determine a "grinding depth" (g) using Eq. (1) (Pickering et al., 2020).

$$g = r_B - \sqrt{r_B^2 - r_{BP}^2} \tag{1}$$

For ellipsoid grains polished less than halfway (e.g., Fig. 1ci), the length and width of the polished face ($L_P$ and $W_P$) must also be measured to estimate the amount removed. In practice, $L_P$ and $W_P$ are often indistinguishable from $L_1$ and $W_1$. For tetragonal grains, the number of pyramidal terminations ($N_p$) must be noted in all cases.





**Table 1: Relationship between 2D grain measurements and geometric values used to calculate volume and surface area.**

| Geometric Classification | Polished Perpendicular to c-Axis | Polished Parallel to c-Axis > Halfway | Polished Parallel to c-axis < Halfway | |
|---|---|---|---|---|
| Ellipsoid | $a = \dfrac{L_1 + W_2}{4}$ <br><br> $b = \dfrac{W_1}{2}$ <br><br> $c = L_2$ | $a = \dfrac{W_1}{2}$ <br><br> $b = W_2$ <br><br> $c = \dfrac{L_1 + L_2}{4}$ | $a = \dfrac{W_1}{2}$ <br><br> $b = \dfrac{W_2 + g}{2}$ <br><br> $c = \dfrac{L_1 + L_2}{4}$ | $a_p = \dfrac{W_p}{2}$ <br><br> $b_p = g$ <br><br> $c_p = \dfrac{L_p}{2}$ |
| Tetragon | $a = \min(W_1, W_2)$ <br><br> $b = \max(W_1, W_2)$ <br><br> $c = \dfrac{L_1 + L_2}{2}$ | $a = 2 \times \min(W_1, W_2)$ <br><br> $b = \max(W_1, W_2)$ <br><br> $c = \dfrac{L_1 + L_2}{2}$ | $a = \min(W_1, W_2) + g$ <br><br> $b = \max(W_1, W_2)$ <br><br> $c = \dfrac{L_1 + L_2}{2}$ | |
| Cylinder | $r = \dfrac{(W_1 + W_2)}{4}$ <br><br> $h = \dfrac{(L_1 + L_2)}{2}$ | $r = \min(W_1, W_2)$ <br><br> $h = \dfrac{(L_1 + L_2)}{2}$ | $r = \dfrac{2r_0 - g}{2}$ <br><br> $h = \dfrac{(L_1 + L_2)}{2}$ | |

**3 Geometry-derived values: Volume, surface area, $R_{SV}$, and mass**

**3.1 Volume and surface area**

**3.1.1 Ellipsoidal Grains**

The ellipsoid semi-axes a, b, and c and polished surface semi-axes $a_P$, $b_P$, and $c_P$ are related to 2D grain measurements $L_1$, $L_2$, $W_1$, $W_2$, $L_P$, $W_P$, and g, as given in Table 1 for each polishing orientation and depth. The ellipsoid coefficient (p) is 1.6075 (Ketcham et al., 2011). When the grain is polished perpendicular to the c-axis or polished parallel to the c-axis and more than halfway ($g > (W_2 + g)/2$) V and SA are calculated using Eq. (2) and (3).

$$V = \frac{2}{3}\pi abc \tag{2}$$

$$SA = 2\pi \left(\frac{a^p b^p + b^p c^p + a^p c^p}{3}\right)^{1/p} \tag{3}$$

When the grain is polished parallel to the c-axis and less than halfway ($g < (W_2 + g)/2$) V and SA are calculated using Eq. (4) and (5).

$$V = \frac{4}{3}\pi abc - \frac{2}{3}\pi a_P b_P c_P \tag{4}$$



$$SA = 4\pi \left(\frac{a^p b^p + b^p c^p + a^p c^p}{3}\right)^{1/p} - 2\pi \left(\frac{ap^p bp^p + bp^p cp^p + ap^p cp^p}{3}\right)^{1/p} \tag{5}$$

V uncertainty is applied as $1\sigma = 21$ % following recommendations in Zeigler et al. (2024). SA uncertainty is unquantified.

### 3.1.2 Tetragonal Grains

The tetragon axes a, b, and c are related to 2D grain measurements $L_1$, $L_2$, $W_1$, $W_2$, and g, as given in Table 1 for each polishing orientation and depth. $N_p$ is the number of pyramidal terminations (0, 1, or 2). When the grain is polished perpendicular to the c-axis V and SA are calculated using Eq. (6) and (7).

$$V = abc - N_p \left(\frac{a}{4}\right)\frac{a^2 + b^2}{3} \tag{6}$$

$$SA = 2(ab + bc + ac) - N_p \left(\frac{a^2 - b^2}{2} + (2 - \sqrt{2})ab\right) - ab \tag{7}$$

When the grain is polished parallel to the c-axis and more than halfway $(g > (a + g)/2)$ V and SA are calculated using Eq. (8) and (9).

$$V = \frac{2abc - N_p \frac{a}{2}\sqrt{b^2 + \frac{4a^2}{3}}}{2} \tag{8}$$

$$SA = \frac{2(2ab + bc + 2ac) - N_p \left(\frac{4a^2 - b^2}{2} + 2ab(2 - \sqrt{2})\right)}{2} \tag{9}$$

When the grain is polished parallel to the c-axis and less than halfway $(g < (a + g)/2)$ V and SA are calculated using Eq. (11) and (12). $g_c$ is an intermediate value that reflects the original whole-grain dimensions (Eq. 10).

$$g_c = c - 2\sqrt{\left(\frac{a}{2}\right)^2 + \left(\frac{b}{2}\right)^2} \tag{10}$$

$$V = (a + g)bc - N_p \frac{a+g}{4}\sqrt{b^2 + \frac{(a+g)^2}{3}} - \frac{1}{2}\left(2gbg_c - N_p \frac{g}{2}\sqrt{b^2 + \frac{4g^2}{3}}\right) \tag{11}$$

$$SA = 2\left((a + g)b + bc + (a + g)c\right) - N_p \left[\frac{(a + g)^2 + b^2}{2} + (a + g)b(2 - \sqrt{2})\right] -$$

$$\frac{1}{2}\left[2(2gb + bg_c + 2gg_c) - N_p \left(\frac{4g^2 + b^2}{2} + 2gb(2 - \sqrt{2})\right)\right] \tag{12}$$

V uncertainty is applied as $1\sigma = 13$ % following recommendations in Zeigler et al. (2024). SA uncertainty is unquantified.

### 3.1.3 Cylindrical Grains

Cylinder radius (r) and height (h) are related to 2D grain measurements $L_1$, $L_2$, $W_1$, $W_2$, and g, as given in Table 1 for
different polishing orientations and depths. When the grain is polished perpendicular to the c-axis V and SA are calculated using Eq. (13) and (14).

$$V = \pi r^2 h \tag{13}$$

$$SA = 2\pi rh + \pi r^2 \tag{14}$$



When the cylinder is polished parallel to the c-axis, the original radius pre-polishing ($r_0$) is estimated using Eq. (15). These

calculations for V and SA also make use of an intermediate value $k$ (Eq. (16) or (19) depending on polishing depth) that

combines $r_0$ and r.

$$r_0 = \frac{(\min(W_1,W_2)+g)+\max(W_1,W_2)}{4} \tag{15}$$

When the grain is polished parallel to the c-axis and more than halfway ($g > r_0$) $k$ is calculated using Eq. (16) and V and SA

are calculated using Eq. (17) and (18).

$$k = \sqrt{r_0^2 - (r_0 - r)^2} \tag{16}$$

$$V = \frac{1}{2}\pi rhk \tag{17}$$

$$SA = \pi rk + \frac{\pi h}{2}\left(3(r + k) - \sqrt{(3r + k)(r + 3k)}\right) \tag{18}$$

When the grain is polished parallel to the c-axis and less than halfway ($g < r_0$) $k$ is calculated using Eq. (19) and V and SA

are calculated using Eq. (20) and (21).

$$k = \sqrt{r_0^2 - (r_0 - g)^2} \tag{19}$$

$$V = \pi h\left(r_0^2 - \frac{1}{2}gk\right) \tag{20}$$

$$SA = 2\pi\left(r_0^2 - \frac{1}{2}gk\right) + h\left(\pi(2r_0 - g) - \frac{\pi}{2}\left(3(g + k) - \sqrt{(3g + k)(3k + g)}\right)\right) \tag{21}$$

Volume uncertainty has not been quantified for cylindrical grains like it has for other geometries (Cooperdock et al., 2019;

Zeigler et al., 2023, 2024). In the absence of this, uncertainty on cylindrical V is applied as $1\sigma = 21$ % (the largest quantified

uncertainty for zircon, Zeigler et al., 2024) as a conservative estimate. Future work should establish a quantitative V

uncertainty value and correction for cylinders as this is a common geometry for abraded grains. SA uncertainty is

unquantified.

### 3.2 $R_{SV}$

$R_{SV}$ is calculated using Eq. (3) from Ketcham et al. (2011), notated Eq. (22) below.

$$R_{SV} = \frac{3V}{SA} \tag{22}$$

$R_{SV}$ inherits uncertainty from V.

### 3.3 Mass

M is calculated assuming an average zircon density 4.65 x $10^{-12}$ g µm$^{-3}$ using Eq. (23).

$$M = (4.65 \times 10^{-12})V \tag{23}$$

M inherits uncertainty from V.



## 4 Geometry and analytical measurement-derived values: Parent isotope concentrations, eU, $F_T$, and $R_{FT}$

### 4.1 Parent isotope concentrations

Parent isotope concentrations for uranium (U), thorium (Th), and samarium (Sm) in ppm are calculated from parent isotope masses in ng and total M using Eq. (24).

$$[X] = \frac{(ng\,X)/1000}{M} \tag{24}$$

Parent isotope concentration uncertainty is propagated from the total analytical uncertainty on the ng measurements and M uncertainty inherited from V.

### 4.2 eU

eU is calculated using Eq. (A9) from Cooperdock et al. (2019), Eq. (25) below.

$$eU = [U] + 0.238[Th] + 0.0083[^{147}Sm] \tag{25}$$

Uncertainty on eU is propagated from uncertainties on the U, Th, and Sm concentrations combining total analytical uncertainty and V uncertainty.

### 4.3 $F_T$

$F_T$ values are calculated using the weighted mean stopping distances $S_x$ of an alpha particle for a given parent isotope decay chain (15.55, 18.05, 18.43, and 4.76 µm for $^{238}$U, $^{235}$U, $^{232}$Th, and $^{147}$Sm, respectively, Ketcham et al., 2011), $R_{SV}$ dependent on the crystal volume and ejection surface area (Eq. (22)). and geometry-specific $F_T$ equations from Ketcham et al. (2011) (Eq. (26), (27), and (28) below). The relationship between a, b, c, r, and h and 2D measurements varies by geometry, polishing orientation, and polishing depth as given in Table 1.

#### 4.3.1 Ellipsoidal Grains

$$F_{T,x} = 1 - \frac{3}{4}\left(\frac{S_x}{R_{SV}}\right) + \left[\frac{1}{16} + 0.1686\left(1 - \frac{a}{R_{SV}}\right)^2\right]\left(\frac{S_x}{R_{SV}}\right)^3 \tag{26}$$

#### 4.3.2 Tetragonal Grains

$$F_{T,x} = 1 - \frac{3}{4}\left(\frac{S_x}{R_{SV}}\right) + \frac{0.2095(a+b+c)S_x^2}{abc} - 0.00995\frac{S_x^3}{abc} \tag{27}$$

#### 4.3.3 Cylindrical Grains

$$F_{T,x} = 1 - \frac{1}{2}\frac{(r+h)S_x}{rh} + 0.2122\frac{S_x^2}{rh} + 0.0153\frac{S_x^3}{r^3} \tag{28}$$

Isotope-specific $F_T$ uncertainties are applied following recommendations in Zeigler et al. (2024) for ellipsoid and tetragonal grain geometries; for cylindrical geometries the larger of the recommended uncertainties for the other geometries is applied.





Ellipsoid: 3 %, 4 %, 4 %, and 1 % for $F_{T,238}$, $F_{T,235}$, $F_{T,232}$, and $F_{T,147}$, respectively. Tetragonal/Cylindrical: 3 %, 4 %, 5 %, and 1 % for $F_{T,238}$, $F_{T,235}$, $F_{T,232}$, and $F_{T,147}$, respectively.

### 4.3.4 Combined $F_T$

Combined $F_T$ is calculated for each grain using equations from Cooperdock et al. (2019) and activities for $^{238}$U and $^{232}$Th, $A_{238}$ and $A_{232}$, respectively (Eq. (29), (30), and (31)).

$$A_{238} = (1.04 + 0.247[\text{Th/U}])^{-1} \qquad (29)$$

$$A_{232} = (1 + 4.21[\text{Th/U}])^{-1} \qquad (30)$$

$$\overline{F_T} = A_{238}F_{T,238} + A_{232}F_{T,232} + (1 - A_{238} - A_{232})F_{T,235} \qquad (31)$$

Combined $F_T$ uncertainty is propagated from the uncertainties on the U and Th concentrations and isotope-specific $F_T$ values.

### 4.4 $R_{FT}$

$R_{FT}$ is calculated using Eq. (6) from Cooperdock et al. (2019) and related equations (Eq. (32), (33), (34)).

$$S/R = 1.681 - 2.428\overline{F_T} + 1.153\overline{F_T}^2 - 0.406\overline{F_T}^3 \qquad (32)$$

$$\bar{S} = A_{238}S_{238} + A_{232}S_{232} + (1 - A_{238} - A_{232})S_{235} \qquad (33)$$

$$R_{FT} = \frac{\bar{S}}{S/R} \qquad (34)$$

Uncertainty on $R_{FT}$ is applied as $1\sigma = 8$ % $R_{FT}$ following recommendations in Zeigler et al. (2024).

### 5 Comparison with other methods

In this section, I compare eU and combined $F_T$ values calculated using the methods presented here with alternative methods. First, I compare eU and $F_T$ with the uncorrected method of Ketcham et al. (2011) on which the method presented here is
based (Fig. 2a, b). I also compare $F_T$ calculated using the method of Reiners et al. (2007) (Fig. 2c). Comparisons were made using a detrital zircon dataset (n = 70) generated at the University of Colorado Thermochronology Research and Instrumentation Lab (Table S1).

Comparison among methods using this real dataset show a mixed outcome. While values for eU and $F_T$ are generally similar
regardless of method, there are significant differences for individual grains, particularly grains with tetragonal and cylindrical geometries (Fig. 2). The eU calculated using the uncorrected Ketcham et al. method is typically within the $1\sigma$ uncertainty of eU calculated using methods in this study (Fig. 2a), with a median percent difference of 3 %. However, there are exceptions for cylindrical grains and grains with low U concentrations and the percent difference for eU can be as high as 92 %. The percent differences between $F_T$ calculated using the uncorrected Ketcham et al. method and Reiners et al. method
and $F_T$ calculated in this study can be up to 31 and 58 %, respectively, with median percent differences of 6 and 3 %. The





uncorrected Ketcham et al. method generally underpredicts $F_T$ (Fig. 2b) while the Reiners et al. method generally overpredicts $F_T$ (Fig. 2c).

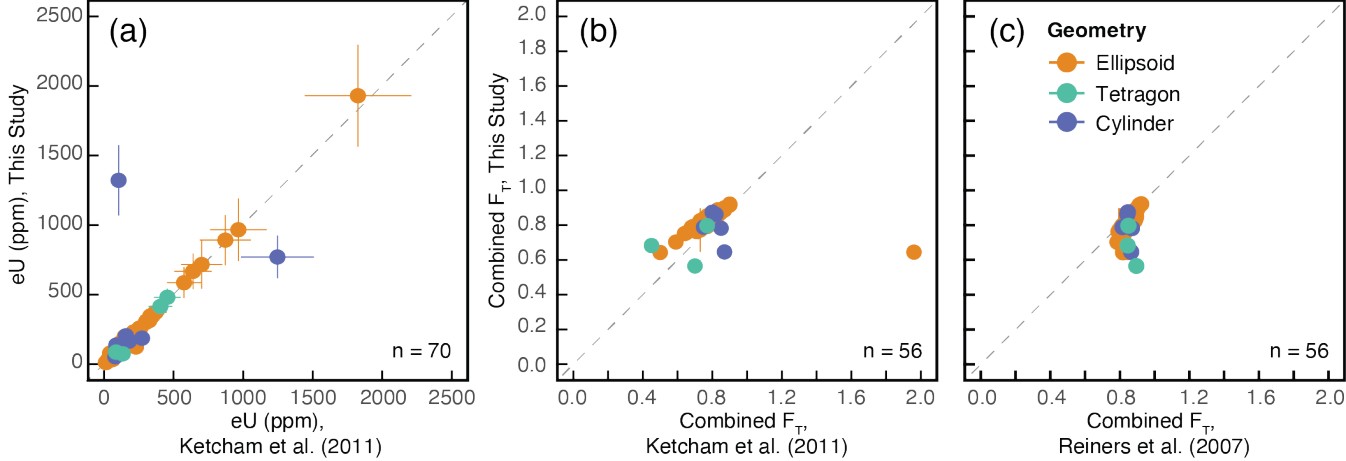

Figure 02 **Comparison of values calculated using the methods described in this study and alternatives used in the literature. (a) eU comparison between this study and the method of Ketcham et al. (2011) with no correction for polishing. (b) Combined $F_T$ comparison between this study and the method of Ketcham et al. (2011) with no correction for polishing. (c) Combined $F_T$ comparison between this study and the method of Reiners et al. (2007). Only grains which are not interior fragments (n = 56) are plotted for $F_T$ comparison. All grains (n = 70) plotted for eU comparison. Dashed gray line represents parity in all panels. Color key corresponding to idealized original grain geometry is the same in all panels.**

Although the differences between methods is generally small, the presence of much larger differences for a subset of grains could heavily bias interpretations when considered in the context of typical thermochronology studies. Most thermochronology studies consist of 4-10 grains analysed per sample, such that inaccurate calculation of volume and derived values for just one grain can have a large impact on interpretation of the entire dataset. In situ grain characterization is in part motivated by the need for full understanding of grains given small sample sizes, so it is important that the positives gained by polishing and additional analyses not be negated by inaccurate volume calculation. Based on the comparisons shown here, I recommend against approaches that ignore the effect of polishing altogether (e.g., the uncorrected Ketcham et al. method), as this can result in discrepancies larger than 5 % for both eU and $F_T$. The Reiners et al. method of accounting for $F_T$ corrections due to polishing, while frequently resulting in < 5 % difference, can also vary more significantly, likely due to simplifying assumptions made by this method regarding grain geometry, orientation, and depth of polishing. The method presented here makes fewer assumptions and is therefore preferred, although full comparison with volumes obtained via 3D imaging like the comparisons done in Zeigler et al. (2024) are necessary to determine the true accuracy of the values obtained via this method.

## 6 Conclusions

In order to extract the maximum amount of useful data from individual minerals used to obtain (U-Th)/He dates, methods for calculating grain V and data derived from V must be able to account for the impact of polishing grains for in situ analyses. Previous methods have addressed corrections for some, but not all derived data. In particular, data related to parent isotope concentrations have previously been ignored. The method presented here provides a means to obtain V, SA and all data derived from these values as directly as possible regardless of original grain geometry and polishing conditions. Although in

many cases the discrepancy between this method and existing approximations is small (< 5 percent difference), values can differ by as much as 92 percent difference for individual grains. The small sample size of most (U-Th)/He datasets makes accurate representation especially important, as an incorrect value in this small sample size can lead to significant misinterpretation. The equations presented here are available in the accompanying R script (Code Availability) which should make implementation of this method for accounting for polishing relatively straight forward, opening up new possibilities

for in situ data collection to accompany conventional zircon (U-Th)/He thermochronology.

**Code and data availability**

The R script to calculate all values can be accessed via GitHub (https://github.com/Barra-Peak/polished-ZHe-derived-values). The script outputs two .xlsx files: one summarizing all derived values, and one formatted for use with the HeCalc python program to calculate corrected (U-Th)/He dates and uncertainty (Martin et al., 2022; 2023). All data used in the

method comparison and plotted in Fig. 2 can be found in the Supplement.

**Author contributions**

BAP conceptualized the study, developed the equations, code, grain measurement protocol, prepared the test dataset and method comparison, and prepared the manuscript.

**Competing interests**

The author declares no conflict of interest.

**Acknowledgements**

Thank you to Spencer Zeigler for proofreading the R script and making suggestions to improve its efficiency. Thank you to Rebecca Flowers for suggestions that improved the clarity of early versions of the manuscript.





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
