# Peer review of "Technical Note: Improved calculation of volume, FT correction, and other derived data for polished zircon (U-Th)/He thermochronology"

_Geochronology, 2024_

## Author Comment (AC1)

**Response to RC1 Comments**

Barra Peak
Jan. 20, 2025

I thank the reviewer for their comments and feedback on the manuscript. Most of the comments deal with clarifying and providing more explanation of the overall approach and using more explicit, less ambiguous terms. I see no problem implementing these changes and believe they will greatly improve the manuscript. I think it likely that many readers of the original manuscript will have similar questions and comments as RC1 and am providing responses here to help clarify prior to the end of open discussion. I will provide additional responses after incorporating other feedback.

RC1: From what I understand from your manuscript, your proposed approach will likely under- or overestimate the Ft value since you are using volume and surface area calculation of the polished grains for your calculation of Ft. The correct Ft value is dependent on both the original grain geometry and the resulting after polishing and only if half a grain is polished both will be similar. The resulting Ft value calculated with your approach will be smaller than the corresponding Ft value of the whole grain, in case polishing removes less than half of the grain. The opposite is in case more than half the grain is removed during polishing. Either show the difference to the correct value and state the limitation or implement the correct calculation.

AC: My approach is intended to provide an $F_T$ correction for the grain fragment remaining after polishing rather than the whole original grain, regardless of how much of the grain remains. Since only the fragment is analyzed for conventional bulk U, Th, Sm, and He measurements, applying an $F_T$ correction based on the original, unpolished grain volume and surface area would be inappropriate without also somehow correcting these isotopic measurements to reflect the original whole grain morphology. Although this is theoretically mathematically feasible, the uncertainty associated with doing this – both quantifiable and unquantifiable – make it a worse option in my opinion than proceeding with just the direct measurements of the remaining fragment as done here.

I can see that this basis of the approach presented – calculating values for the polished grain fragments, not the original whole grains – was not communicated effectively in the original manuscript and I will emphasize this explicitly in revisions.

RC1: Line 1,16: Specify what you mean with 'derived value calculation'. Also, later you often say something like 'other values'. Please make sure that you always specify the measurements you are referring to.

AC: Thank you for pointing out that "derived value" is vague and should be avoided or better defined. Here, I am using "derived values" in a manner similar to Flowers et al., (2022) to refer to data or correction values such as $F_T$ that are not directly measured but are derived from direct

measurements, typically during "data reduction" steps of (U-Th)/He thermochronology workflows. Where appropriate, I will reference just the specific values being discussed. It is still sometimes necessary to refer to this category of values derived from direct measurements as a whole, but I will add a sentence explicitly defining this categorization.

RC1: Line 8-17: You state that the proposed protocol is beneficial for in situ measurements (line 14) and later for bulk grain (e.g. line 16). Please state clearly which method (in-situ and/or bulk grain) would benefit.

AC: The protocol is beneficial for bulk grain conventional (U-Th)/He thermochronology applications in which additional in-situ measurements on the same grains are also required/desired. Examples of complimentary in-situ data collection can include LA-ICPMS or SIMS U-Pb dating or trace element analysis, cathodoluminescence or backscatter electron imagery, etc. I will reword sentences in the abstract and introduction to clarify cases in which the protocol presented is helpful.

RC1: Line 49-50: You may want to reference to my approach using a set of orthogonal microscopic pictures to derive whole-grain geometries (Glotzbach et al. 2019 – Chemical Geology).

AC: Thank you for the suggestion; apologies for the oversight.

RC1: Line 48-58: Please clearly state what your approach is. Measuring only after mounting/polishing or, what I guess, two measurements are required before and after mounting/polishing.

AC: The approach uses one set of measurements made after the grains are polished. This will be explained explicitly in the Introduction and Section 2. Measuring once after grains are polished adds minimally to sample preparation time and avoids introducing additional opportunities for grain selection bias, which can be especially problematic in some applications such as detrital zircon provenance analysis.

RC1: Line 72: The word 'irrelevant' is somewhat misleading here, since the depth to which grains are polished is impacted V, SA and other related parameters and is not irrelevant. I guess you mean that it is easy to account/correct for.

AC: I agree better wording is needed here. In some cases, polishing depth is not needed since the volume and surface area of the remaining fragment can be easily calculated from measurements of the fragment alone. In other cases this is not possible and polishing depth is needed.

RC1: Line 74-77: See above, in case you measure individual grains before and after mounting/polishing this would not be required. Therefore I guess you are measuring only after mounting/polishing and derive the depth of polishing from the glass beads.

AC: Yes, grains are only measured after polishing. The Abstract, Introduction, and Section 2 will be revised to state this explicitly.

RC1: Line 85-90: I do not fully understand how you can estimate the correct values of a and b for an ellipsoid (r for cylinder) in case more than half of the grain is polished away. The equations that you state will be minimum values for a and b (e.g. b=W1/2).

AC: Thank you for this feedback, which illuminates that a key point about how the method is applied has not been communicated effectively in the original manuscript. The classification of grains as either ellipsoid, tetragon, or cylindrical geometries relates to the original geometry of the whole grain, prior to polishing. However, the calculation of volume and surface area relates just to the grain fragment remaining after polishing. Volume and surface area calculations are related to the original grain geometry in the sense that the geometry of the polished fragment reflects this original geometry, but do not necessarily use values a, b, c, (ellipsoids and tetragons) or r and h (cylinders) related to a, b, c, r, or h, of the original geometry – these values are defined for the fragment. This clarification will be added to Section 2 and elsewhere as needed.

To address the specific points raised in the comment:

In the case of an originally ellipsoid grain polished more than half away, the geometry of the remaining fragment is approximated as half an ellipsoid with semi-axes a and b determined using measurements of the *remaining fragment*. The semi-axes of the original whole grain and polishing depth are not used. I agree language clarifying that this is an approximation should be added.

In the case of grains that are originally cylinders polished parallel to the c-axis, I also agree that the approximation for r and subsequent volume and surface area derivations presented in the original manuscript are not the best approximations possible. An explanation of a better approach that does not attempt to tie volume and surface area of these grain fragments to equations for a cylinder but instead treats them more simply as prisms with half-ellipse cross sections is included at the end of this reply. This updated approach for cylindrical grains will replace the existing approach in the revised manuscript and R script.

RC1: Line 86: Specify what the ellipsoid coefficient is.

AC: The ellipsoid coefficient is a feature of Knud Thomsen's formula to approximate ellipsoid surface area employed in Ketcham et al., (2011) and adapted here. I will add a fuller citation of this value in the revised manuscript.

RC1: Line 114-115: Same as for the ellipsoid does apply for a cylinder, it is not possible to correctly determine r when more than half the grain is gone. The equation that you are using r= min(W1,W2) will underestimate r. Why not using equation 1 to estimate the correct radius?

AC: I agree this approach is not optimal. A better approach is described at the end of these replies as in the previous response to the Line 85-90 comment.

RC1: Line 158-186: It is unclear to me if you calculate the Ft for the whole grain or the mounted/polished grain. The Ft value of mounted/polished grains will in most cases be

higher/lower than the theoretical value of the whole grain (similar only if exactly half the grain is removed).

AC: $F_T$ is calculated for the polished grain fragment only. The manuscript will be revised to explicitly state this.

RC1: Line 187: Please add more details on how you did the comparison, are this read data or synthetic data and give details how the methods of Ketcham and Reiners differ from your approach.

AC: Thank you for the feedback that more details are needed. The data used is real dataset acquired for a detrital zircon sample. I agree real vs. synthetic data is an important distinction that should be explicit. The Ketcham method is already described in Section 2 in contrast to my approach. I will clarify Section 2 to more clearly differentiate between them and reference that description in Section 5. Adding details about the Reiners method is not a problem.

RC1: Line 189: Please clarify what you mean with 'uncorrected method'?

AC: Thank you for the feedback that this is unclear. The use of "uncorrected method' in this context was meant to differentiate the method of Ketcham et al. (2011) for calculating volume, surface area, and $F_T$ for whole grains from my protocol for calculating these values for polished grain fragments. As stated throughout the text, my protocol draws heavily on the approach and equations employed by Ketcham et al. but modifies them so that they are correct for polished fragments. However, based on this and other comments, this wording has only created confusion. The revised manuscript will refer simply to the "Ketcham et al. method" in contrast to the method presented in the manuscript.

**Updated volume and surface area for cylindrical grains polished parallel to c-axis**

For grains that are originally cylindrical and polished parallel to the c-axis, the removed portion of the grain or grain fragment remaining, depending on degree of polishing, can be approximated as prisms with cross sections perpendicular to the c-axis that are half ellipses (see diagrams in Table 1 below). Using this approximation, the volume and surface area of the remaining polished grain fragments are calculated using the area of an ellipse ($A_{ellipse}$, Eq 1) and Ramanujan's Formula for the approximate perimeter of an ellipse ($P_{ellipse}$, Eq 2). Updated equations for the volume (V) and surface area (SA) of originally cylindrical grains incorporating Eq 1 and 2 are given as Eq 4 – 7 in Table 1 below.

$$A_{ellipse} = \pi ab \tag{1}$$

$$P_{ellipse} = \pi(a + b)\left[1 + \frac{3k}{10 + \sqrt{4 - 3h}}\right] \tag{2}$$

Where $a$ and $b$ are the ellipse semi-axes and $k$ is defined as $k = \frac{(a-b)^2}{(a+b)^2}$.

To determine the degree of polishing using the polishing depth $g$ (calculated as described in the manuscript using measurements of polished glass beads), the original radius of the cylindrical grain before polishing is calculated using Equation 3.

$$r = \frac{g + W_2}{2} \tag{3}$$

**Table 1** Originally cylindrical grains polished parallel to c-axis volume and surface area calculation. Gray segments in cross sectional diagrams represent grain fragment remaining after polishing for which volume and $F_T$ surface area are calculated.

| Degree of Polishing | Cross Section Perpendicular to c-axis | Values Defined Using 2D Grain Measurements | Volume and Surface Area Equations |
|---|---|---|---|
| $g < r$ Polished <halfway |  | $a = g$ $$b = \sqrt{r^2 - (r-g)^2}$$ $$h = \frac{L_1 + L_2}{2}$$ $$k = \frac{(a-b)^2}{(a+b)^2}$$ | $$V = \left(\pi r^2 - \frac{\pi}{2}ab\right)h \quad (4)$$ $$SA = 2\left(\pi r^2 - \frac{\pi}{2}ab\right) + \left[2\pi - \frac{\pi(a+b)}{2}\left[1 + \frac{3k}{10 + \sqrt{4-3k}}\right]\right]h \quad (5)$$ |
| $g > r$ Polished >halfway |  | $a = W_2$ $$b = \frac{W_1}{2}$$ $$h = \frac{L_1 + L_2}{2}$$ $$k = \frac{(a-b)^2}{(a+b)^2}$$ | $$V = \frac{\pi}{2}abh \quad (6)$$ $$SA = \pi ab + \frac{\pi(a+b)}{2}\left[1 + \frac{3k}{10 + \sqrt{4-3k}}\right]h \quad (7)$$ |

---

## Author Comment (AC2)

**Response to RC2 Comments**

Barra Peak
March 31, 2025

RC2: The author provides a set of equations to calculate volume and surface area, which seems generally uncontroversial, as far as I can tell. The R code may be useful for some. Generally, the manuscript could be improved if it more clearly stated what measurements are actually required as input for each of the geometric cases, and if the equations were appropriately justified when they are introduced. As it is, there's insufficient discussion to justify some of the assumptions going into the equations, and many of the variables+equations are poorly explained or not explained at all.

AC: I appreciate the reviewers taking the time to provide detailed feedback on where they found the manuscript confusing and needing clarification. They have made many suggestions which once incorporated will greatly improve the manuscript. I agree with the majority of their criticisms and need for additional discussion of how the protocols presented in this manuscript differ from existing approaches in order to justify adoption of these protocols.

I am not opposed to including more explanation of the background equations used in the protocol described here and I agree that doing so would make the paper more accessible to a broader audience. However, to me, this feedback suggests that the reviewers and I may have different understandings about the purpose of a technical note paper and clarification of the intended purpose from the editor would be helpful prior to revisions. I agree completely with the need to better explain and justify how my proposed protocol differs from existing methods; however, I am not sure how much background explanation of existing equations and variables on which the method is based is needed. I wrote the original manuscript with the understanding that a technical note should be brief and targeted to a very specific methodological problem or advancement – e.g., something that in other cases might end up in the supplement of a paper framed around a geologic question and interpretation. Thus, my understanding of the intended audience of a technical note is other scientists who run into the same specific methodological problem—other thermochronologists conducting bulk-helium analyses on polished grain fragments. As such, I assumed the audience would have a common background regarding how geometric grain classifications are made and the justification for the use of general geometric equations for volume and surface area to describe these geometries such that rather than providing detailed explanation of existing equations, citation of the relevant literature (e.g., "Eq. 3 of Ketcham et al., 2011") would be sufficient. The fact that both reviewers asked for more detail and discussion on this background shows my assumptions were wrong and suggests to me that they may have a different interpretation of the intent of a technical note. Editorial input about the intended purpose and audience of a technical note as well as clarification about length limits, would help me determine the best course for revisions and if a technical note is actually the best channel to share the methodological protocols presented here.

Other comments:

RC2: First, I would suggest that the author reconsider the use of "volume and derived value" in the title, which could be more informative if it specified, for example, "improved calculation of

volume, FT correction, and other derived values for polished zircon"… At the end of the day, the FT correction is what most readers are interested in.

AC: I agree, thank you for the suggestion.

RC2: Second, the manuscript would be much improved with greater discussion of the specifics of how the proposed calculations differ from previous protocols (e.g. in Lines 210-215). The equations are presented, but not much is done to demonstrate their superiority, besides Figure 2. Which cases lead to the large (95%) variation? It seems to be just a single grain or two? In what cases/geometries/grain sizes generally have minimal difference between the different protocols? Is it small grains that are particularly effected? or when very little of the grain is polished, or a lot of it is polished away?

AC: I agree that this would strengthen the manuscript. Notably, the conditions under which the different protocols are designed to apply are very different, with the Reiners et al. protocol limited to ellipsoid or tetragonal grains polished parallel to the c-axis and less than halfway through the grain. I agree that this is an important point that is missing from the current manuscript. A better Figure 2 can be used to address some of the specific questions raised by the reviewer regarding the impact of grain size and polishing depth rather than extensive additions to the text since it is my understanding that the text is already on the long side for a technical note. Editorial clarification on this would be appreciated.

To more fully assess the differences between protocols in cases when the Reiners et al. protocol is designed to be applicable, a comparison of a synthetic set of zircon data with a range of size, parent isotope concentrations, and polishing depths could be made. I did not include this previously as the conditions that satisfy the Reiners et al. protocol starting conditions apply to only a small subset of possible conditions, which is reflected in the real dataset presented and a main motivator for the development of the new protocol presented. Again, I realize this point is not clearly articulated in the present manuscript and needs to be more explicitly stated.

RC2: A histogram could be provided showing the % difference between the different protocols for each grain in the dataset. Is there a systematic skew towards overestimating or underestimating FT?

AC: I agree this could be a helpful addition to Figure 2; thank you for the suggestion.

RC2: Notably, Fig. 2 only shows that these protocol are different. But it's not immediately clear why, practically speaking, the additional complication of assigning grains to particular sub-classes of geometries and degrees of polishing based on limited 2D measurements from a polished mount wouldn't simply be introducing more assumptions and/or errors. I could certainly imagine how these detailed calculations here could be better - but I don't think that's necessarily the case, and the author needs to demonstrate that.

AC: I agree that further discussion and justification of this method compared to others is necessary. I think this would be done best using a synthetic dataset, similar to what was done in the He and Reiners (2022) paper. I address this further in subsequent responses.

RC2: For example, the author states that "The Reiners et al. method of accounting for FT corrections due to polishing, while frequently resulting in < 5 % difference, can also vary more significantly, likely due to simplifying assumptions made by this method regarding grain geometry, orientation, and depth of polishing." The author should expand on this sentence and explicitly discuss those simplifying assumptions. How and when exactly do they vary so significantly? And most importantly, for many readers, the question is whether it is practical to move beyond those simplifying assumptions. It would be helpful if the author distinguished the specifics of the cases (e.g. the one with the 92% difference) that led to the large difference. Why and how are the approximations that are used here (particularly for the c-axis parallel cases) better/different than the simplifying assumptions used by others?

AC: I agree that more explicit discussion of the assumptions is needed. The Reiners et al. method assumes grains are polished parallel to the c-axis, polished less than halfway through the grain but greater than one alpha-stopping distance, and does not provide a set of corrections for grains with cylindrical geometries. Further, the Reiners et al. method makes simplifying assumptions about the degrees of symmetry in ellipsoidal grains and the number of terminations present for tetragonal grains. These assumptions are not met by the majority of the real grains and polishing conditions presented in the manuscript, suggesting that the Reiners et al. method should not be used to correct these grains. However, I agree the manuscript currently lacks further justification of this.

To fully show the improvement of the protocol presented here over the Reiners et al. method comparison between methods using a synthetic dataset can be added that covers a broader range of conditions than the real dataset.

RC2: Third, when polished perpendicular to the c-axis, the calculations would essentially be the same case as the fragmentation correction for grains with one end broken, which we discussed in a similar paper (He and Reiners, 2022). For these cases, I imagine the modified FT would be exactly the same as the protocol propose here?

AC: Yes, the approach is the same in which the polished fragment has an $F_T$ correction equivalent to a whole crystal with length 2x the fragment length and I apologize for not citing this He and Reiners (2022) in regards to this approach. The revised manuscript will be updated with this citation.

The same approach is used for ellipsoidal and tetragonal grains polished parallel to the c-axis when more than half the original grain is removed except in these cases the relevant plane of symmetry is different and thus the axis modified by a factor of 2 is the b-axis in the case of ellipsoids and the a-axis in the case of tetragons.

RC2: Finally, something additional that would be relevant to add in the discussion: the idea that the SA/V of polished grains can be used to modify FT corrections assumes that the polynomial function relating SA/V to FT is nearly identical for most geometries. But it is not entirely identical, and polished grains would deviate pretty far from ideal geometries used to determine the SA/V-FT functions.

AC: I agree that this is worth discussing. Revisions to Section 2 will reflect this.

Other comments:

RC2: It's not clear from Fig. 1 what the different labels (e.g. w1 wp) are referring to in many of the diagrams.

AC: Thank you for the feedback that this is unclear. W1, W2, L1, and L2 are the measurements made of the grain while a, b, c, r, and h refer to the axes, semi-axes, radius or height of the idealized geometry. The figure is meant to show visually how these relate to each other to compliment the mathematical relationships between measurements and geometric parameters given in Table 1. Sets of W1, L1 and W2, L2 measurements are made by rotating the grains 90° such that these measurements can also define a rectangular prism that surrounds the grain as in the example below. Does the example better confer the intended relationship? Measurements and parameters axes could also be color-coded to their respective labels.

[Figure]

RC2: What you call SA is not actually surface area - but rather something like the alpha-ejection-affected-surface area. I suggest using a subscript to clarify this ($SA_\alpha$), or something similar, as it is can be confusing to readers. Note that in He and Reiners 2022, we called $\beta_\alpha$ =the ratio of alpha-ejection-affected surface area to volume.

AC: This is true, and this modification of "SA" from the true SA is explained in lines 68-69. I will clarify this modification of "SA" earlier in the manuscript.

The protocols proposed in the manuscript are based directly on the protocols of Ketcham et al. (2011) which uses SA to refer to alpha-ejection surface area (albeit assuming the entire grain surface area is an ejection surface) and equivalent spherical radius, $R_S$, defined as 3V/SA in the formulation of $F_T$ rather than $\beta_\alpha$. I adopted the same formulation of $F_T$ and $R_S$ terminology to be consistent with Ketcham et al. and to aid a reader who would likely be comparing directly between that and the protocols presented here ($R_S$ was changed to $R_{SV}$ reflecting reporting standards used at CU TRaIL to differentiate from $R_{FT}$). $R_{SV}$ and $R_{FT}$ are important values in their own right, as they are commonly reported as a description of grain size and used in thermal history modeling software programs. In the interest of not introducing even more variables, I elected to not reference $\beta$ in the original manuscript, but I am happy to add reference to it since more readers may be familiar with it than $R_S$.

RC2: There should be more details about the test dataset: what was the measurement protocol? the range of grain sizes? how were the grains assigned into different geometries if they were already polished?

AC: Grain measurement protocol is partially described in Line 55. Grains fragments were easy to classify post-polishing once removed from the epoxy mount and viewed under a microscope as they looked like either half an ellipsoid, half a cylinder, or half a prismatic crystal with clear faces 90° to each other. These details will be added to the revised manuscript.

---

## Author Response (AR1)

**Response to summary editor and reviewer comments**

EC: The two reviewers were largely in agreement; they acknowledged the importance of the topic addressed in the manuscript, but pointed out that there was room for improvement in the presentation, including the wording, explanations of the equations/parameters, and comparison with the previous protocols. I generally agree with their comments and ask the authors to update their manuscript accordingly. If the two reviewers, who are experts in Ft calculations, felt that the approach was not explained well enough, it is likely that most readers will find it even more so. In response to the reviewer's comments, the authors proposed a comparison with the previous protocols using a synthetic dataset, which could be very useful information for potential users of the R script. Since the manuscript is a Technical Note, I think there is a limit to the number of pages, but I hope that the authors will incorporate this well into the manuscript, if necessary, using a supplement file, etc. - this is a personal expectation and not a requirement for acceptance. To conclude, I believe that this manuscript is potentially suitable for publication in the GChron journal after moderate revisions.

RC2: The author provides a set of equations to calculate volume and surface area, which seems generally uncontroversial, as far as I can tell. The R code may be useful for some. Generally, the manuscript could be improved if it more clearly stated what measurements are actually required as input for each of the geometric cases, and if the equations were appropriately justified when they are introduced. As it is, there's insufficient discussion to justify some of the assumptions going into the equations, and many of the variables + equations are poorly explained or not explained at all.

AC: I am grateful to the reviewers for taking the time to provide detailed feedback on where the manuscript needs improvement. The majority of the comments from the editor and both reviewers request additional clarification about the method presented in the manuscript, both in terms of how minerals are measured and classified, and the overall approach to FT calculation. Both reviewers also requested more details regarding how the new protocol differs (or doesn't) from previously published methods that are used for comparison. I agree with the need for additional clarification, particularly for explaining the general approach to the new protocol. I have substantially reworded and added to the text in order to address these requests and correspondingly reorganized some of the content to keep the total length of the main text appropriate for a technical note. Equations now mainly appear in an Appendix (Appendix A), so that the main text of the manuscript is devoted to explanation of the approach and required measurements (Section 2) and evaluation of the new protocol (new Section 3). The main evaluation of the new protocol (new Section 3) has been updated to include comparisons using a large synthetic dataset. The comparisons using a real dataset that appeared in the original manuscript have also been revised and moved to Supplementary Text. The real dataset confirms results found with the synthetic dataset and now plays only a supporting role in the manuscript, making it appropriate for a supplement.

From the reviewer comments, it was clear that central components of how the new method is applied were not communicated in the original manuscript. Revisions to address specific comments and points of confusion are described below, but in general, the method is designed to calculate volume, alpha-ejection surface area,  $F_T$ , and all other values **after** grinding and polishing of grains has occurred and **only for the remaining grain fragment**. No attempt is made to apply "whole grain" values, as these would not be appropriate to combine with isotopic measurements made on the grain fragments anyway. Grain geometry classification is based on the grain fragment itself, not the original grain (although the two are often related - e.g., an ellipsoid grain typically becomes half an ellipsoid when polished). An expanded explanation of this approach is included in revisions to Section 2 and in Appendix A.

To address requests for more details and discussion of  $F_T$  comparisons between the new and existing protocols, I have reframed the method evaluation section (Section 3) around a new synthetic dataset containing 16128 synthetic polished and whole zircon (Table S1) that covers a much larger range of possible grains than the real dataset originally presented. Additions to Section 2 and Section 3 describe the Ketcham et al. and Reiners et al. methods that are used as points of comparison in more detail, as requested. In the process of making these revisions, I realized the Reiners et al. method had been incorrectly cited, and that has now been fixed.

**Point-by-point reply to editor and reviewer comments**

EC: Table S1: Length P - the long exis... --> Length P - the long axis...

Table S1: Width P... Typically indistinguisable... --> Width P... Typically indistinguishable...

AC: Thank you for pointing out these typos, they have been corrected in the revised tables.

RC1: From what I understand from your manuscript, your proposed approach will likely under- or overestimate the Ft value since you are using volume and surface area calculation of the polished grains for your calculation of Ft. The correct Ft value is dependent on both the original grain geometry and the resulting after polishing and only if half a grain is polished both will be similar. The resulting Ft value calculated with your approach will be smaller than the corresponding Ft value of the whole grain, in case polishing removes less than half of the grain. The opposite is in case more than half the grain is removed during polishing. Either show the difference to the correct value and state the limitation or implement the correct calculation.

AC: My approach is intended to provide an  $F_T$  correction for the grain fragment remaining after polishing rather than the whole original grain, regardless of how much of the grain remains. Since only U, Th, Sm, and He of the fragment are measured, applying an  $F_T$  correction based on the original, unpolished grain volume and surface area would be inappropriate without also somehow correcting these isotopic measurements to reflect the original whole grain morphology. Although this is theoretically mathematically feasible, the uncertainty associated with doing this – both quantifiable and unquantifiable – make it a worse option in my opinion than proceeding with just the direct measurements of the remaining fragment as done here. Based on this and other comments from both reviewers, it is clear to me that the basis of the approach – calculating values for the polished grain fragments, not the original whole grains – was not communicated effectively in the original manuscript. I have revised the Abstract, Introduction, and Section 2 to explicitly state that the protocol I am presenting is for polished grain fragments. I have also revised Figure 1 to illustrate how volume is calculated.

Line 15-18: "To address this issue, this paper presents a comprehensive protocol for calculating volume and alphaejection surface area for polished zircon grain fragments, from which additional data, including eU and FT, are derived. This protocol is implemented after grains have been polished and in situ measurements have been made..."

Line 71-72: "To address the lack of a comprehensive approach to volume-derived data for polished zircon, this contribution presents a protocol and set of equations..."

Line 79-80: "The protocol presented here adapts existing approaches for determining whole-grain V,  $SA_{\alpha}$ , and  $F_{T}$  for ground and polished grain fragments.."

Line 92-96: "In most cases, calculating these values  $[V, SA_{\alpha}, F_T]$  is accomplished by adopting the same approach as He and Reiners (2022) in which the polished grain is treated as a crystal broken along a plane of symmetry such that V and  $SA_{\alpha}$  of the polished fragment are half of a whole "assumed grain" created by reflecting the existing fragment across the plane of polishing (Fig. 1c).  $F_T$  of the fragment is thus equal to  $F_T$  of the assumed grain."

Using the new synthetic dataset, I have also addressed the relationship between  $F_T$  for whole grains and polished grains in a new Figure 2 and related text:

Lines 185-191: "Across all geometries and grain sizes, the majority of grains exhibit expected changes in  $F_T$  with increasing grinding depth: polished  $F_T$  is greater than whole  $F_T$  up to 50 % grinding depth and polished  $F_T$  is less than whole  $F_T$  above 50 %. The smallest ellipsoid grains with the lowest aspect and width ratios are an exception to this pattern, which is likely related to partial removal of the remaining fragment's alpha ejection rim at higher grinding depths. The largest grains exhibit the smallest differences between whole grain and polished grain  $F_T$ , but the difference increases once more than half the grain width is ground away as for other grain sizes (Fig. 2)."

RC1: Line 1,16: Specify what you mean with 'derived value calculation'. Also, later you often say something like 'other values'. Please make sure that you always specify the measurements you are referring to.

AC: Thank you for pointing out that "derived value" is vague and should be avoided or better defined. Here, I am using "derived values" in a manner similar to Flowers et al., (2022a) to refer to data or correction values such as  $F_T$  that are not directly measured but are derived from direct measurements, typically during "data reduction" steps of (U-Th)/He thermochronology workflows. Where appropriate, I have revised the text to specify the value being discussed. It is still sometimes necessary to refer to the category of values derived from direct measurements as a whole, so I have added explanation of "derived data" to the Introduction:

Lines 27 - 29: "((U-Th)/He thermochronology dates and associated data are derived from analytical measurements of parent and daughter isotopes and the volume (V) of the individual mineral grains analysed. These "derived data," include alpha-ejection correction factors (FT), FT-equivalent spherical radius (RFT), effective uranium (eU), and parent isotope concentrations..."

RC1: Line 8-17: You state that the proposed protocol is beneficial for in situ measurements (line 14) and later for bulk grain (e.g. line 16). Please state clearly which method (in-situ and/or bulk grain) would benefit.

AC: The protocol is beneficial for studies in which in-situ and bulk grain conventional (U-Th)/He thermochronology analyses are desired for the same grains. I have revised wording in the Abstract and Introduction to clarify this:

Lines 17-19: "This protocol is implemented after grains have been polished and in situ measurements have been made and can be easily integrated into existing workflows for characterizing and measuring grains for conventional (U-Th)/He analysis."

Lines 40-43: "However, many applications of (U-Th)/He thermochronology, such as detrital zircon applications, benefit from or require mounting and polishing crystals for in situ analyses to characterize chemical zonation, rare-earth element abundances, U-Pb or other geochronology data, etc. prior to (U-Th)/He analysis. Grinding and polishing grains to prepare them for additional analysis removes part of the grain..."

RC1: Line 49-50: You may want to reference to my approach using a set of orthogonal microscopic pictures to derive whole-grain geometries (Glotzbach et al. 2019 – Chemical Geology).

AC: Thank you for the suggestion; I have added this citation.

RC1: Line 48-58: Please clearly state what your approach is. Measuring only after mounting/polishing or, what I guess, two measurements are required before and after mounting/polishing.

AC: The approach uses one set of measurements made after the grains are polished. Section 2 has been revised to make this explicit:

Lines 79-81: "The protocol presented here adapts existing approaches for determining whole-grain V,  $SA_{\alpha}$ , and  $F_T$  for ground and polished grain fragments. First, the polished grains are removed from the mounting medium and inspected and measured..."

RC1: Line 72: The word 'irrelevant' is somewhat misleading here, since the depth to which grains are polished is impacted V, SA and other related parameters and is not irrelevant. I guess you mean that it is easy to account/correct for.

AC: I agree better wording is needed here. In some cases, polishing depth is not needed since the volume and surface area of the remaining fragment can be calculated from measurements of the fragment alone. In other cases this is not possible and polishing depth is needed. I have revised Section 2 to better explain the different cases and updated Figure 1 to show how grinding depth is needed (or not) for calculation of geometric parameters.

Line 92-102: "In most cases, calculating these values is accomplished by adopting the same approach as He and Reiners (2022) in which the polished grain is treated as a crystal broken along a plane of symmetry such that V and

 $SA_{\alpha}$  of the polished fragment are half of a whole "assumed grain" created by reflecting the existing fragment across the plane of polishing (Fig. 1c).  $F_T$  of the fragment is thus equal to  $F_T$  of the assumed grain. This is the approach used for all grains polished perpendicular to the c-axis or parallel to the c-axis and greater than halfway through the original c-axis perpendicular width of the grain (Fig 1c), which can be determined by visual inspection of the polished grain and does not require measurements of thickness pre-polishing. For grains polished parallel to the c-axis and less than halfway through the original width of the grain (again, determined by visual inspection of the grain post-polishing), a different approach to determining V and  $SA_{\alpha}$  is used. In these cases, the original whole grain dimensions are estimated by combining the grain measurements with the grinding depth (g) determined by measurements of spherical glass beads mounted and polished alongside the grains."

RC1: Line 74-77: See above, in case you measure individual grains before and after mounting/polishing this would not be required. Therefore I guess you are measuring only after mounting/polishing and derive the depth of polishing from the glass beads.

AC: Yes, grains are only measured after polishing. As noted above, Section 2 has been revised to clarify this.

RC1: Line 85-90: I do not fully understand how you can estimate the correct values of a and b for an ellipsoid (r for cylinder) in case more than half of the grain is polished away. The equations that you state will be minimum values for a and b (e.g. b=W1/2).

AC: The geometric values calculated are not intended to reflect the original whole grain (whole ellipsoid, etc.), rather, they reflect the remaining fragment, which in most cases is well-approximated by a half ellipsoid, half cylinder, etc. (Figure 1, Lines 90 - 109). The semi-axes of the original whole grain and polishing depth are not used, which is reflected in revisions to Figure 1. Only in cases where the fragment morphology cannot be approximated as half an ideal geometry are the original grain dimensions estimated. To do this, grinding depth is used. I hope that this confusion is alleviated by revisions to clarify that the volume, alpha ejection surface area,  $F_T$  and other values calculated are only for the polished grain fragment, not the original whole grain.

Lines 92-111: "In most cases, calculating these values is accomplished by adopting the same approach as He and Reiners (2022) in which the polished grain is treated as a crystal broken along a plane of symmetry such that V and  $SA_{\alpha}$  of the polished fragment are half of a whole "assumed grain" created by reflecting the existing fragment across the plane of polishing (Fig. 1c).  $F_T$  of the fragment is thus equal to  $F_T$  of the assumed grain. This is the approach used for all grains polished perpendicular to the c-axis or parallel to the c-axis and greater than halfway through the original c-axis perpendicular width of the grain (Fig 1c), which can be determined by visual inspection of the polished grain and does not require measurements of thickness pre-polishing. For grains polished parallel to the c-axis and less than halfway through the original width of the grain (again, determined by visual inspection of the grain post-polishing), a different approach to determining V and  $SA_{\alpha}$  is used. In these cases, the original whole grain dimensions are estimated by combining the grain measurements with the grinding depth (g) determined by measurements of spherical glass beads mounted and polished alongside the grains. Polishing depth is calculated using Eq. (1) (Pickering et al., 2020) and measurements of the radius of the polished bead surface ( $r_{BP}$ ) relative to the full bead radius ( $r_{B}$ ).

$$g = r_B - \sqrt{r_B^2 - r_{BP}^2} \tag{1}$$

Uncertainty on g can be determined through duplicate measurements of multiple embedded beads scattered throughout the grain mount. The estimated whole grain dimensions are used to estimate V and  $SA_{\alpha}$  for the whole original grain. To calculate V and  $SA_{\alpha}$  of the remaining fragment, the V and  $SA_{\alpha}$  of the removed portion of the grain are also estimated and subtracted from the estimated whole. V and  $SA_{\alpha}$  of the removed portion are determined by treating removed portions of the crystals as half crystals of a whole assumed grain in the same manner as grains polished parallel to the c-axis and more than halfway through the original grain width (Fig. 1c). This calculation requires additional measurements of the polished grain surface: length ( $L_P$ ) and width ( $W_P$ ) of the polished face. In practice,  $L_P$  and  $W_P$  are often indistinguishable from  $L_1$  and  $W_1$ ."

In the case of grains that are originally cylinders polished parallel to the c-axis, I agree that the approximation for r and subsequent volume and surface area derivations presented in the original manuscript are not the best approximations possible. I have updated the approach to treat "cylinders" as prisms with ellipsoid pinacoid

terminations (or partial ellipsoids in the case of grains polished parallel to the c-axis, Fig. 1c) that can be described by semi-axes a and b.

**Lines 294-317: "A2. Cylinder volume and alpha-ejection surface area**

"Cylinders" can more accurately be represented as prisms with height (h) and ellipsoidal, rather than circular pinacoid terminations with semi-axes a and b (Fig. 1c). V and  $SA_{\alpha}$  are calculated using the area of an ellipse ( $\pi ab$ ) and Ramanujan's Formula for the perimeter of an ellipse (Eq. A5).

$$P_{ellipse} = \pi(a+b) \left[ 1 + \frac{3k}{10 + \sqrt{4-3k}} \right]$$
 (A5)

Semi-axes of the ellipsoid cross section are denoted as a and b, k is defined as  $(a - b)^2/(a + b)^2$ , h is the height or length of the cylinder. Additionally, r is the original radius of the cylinder pre-polishing reconstructed as  $r = \frac{(g+W_2+W_1)}{4}$ . The semi-axes are related to the 2D grain measurements and g depending on degree of polishing as given in Table 1. When the grain is polished perpendicular to the c-axis the grain is treated as half a symmetric prism broken perpendicular to the c-axis and V and  $SA_{\alpha}$  are calculated using Eq. (A6) and (A7).

$$V = \pi abh \tag{A6}$$

$$SA_{\alpha} = \pi ab + \pi (a+b) \left( 1 + \frac{3k}{10 + \sqrt{4-3k}} \right) h$$
 (A7)

When the grain is polished parallel to the c-axis and more than halfway through the original width  $(g > (W_2 + g)/2)$ , V and  $SA_\alpha$  are calculated using Eq. (A8) and (A9) which treat the fragment as half of a cylinder:

$$V = \frac{1}{2}\pi abh \tag{A8}$$

$$SA_{\alpha} = \pi ab + \frac{\pi(a+b)}{2} \left( 1 + \frac{3k}{10 + \sqrt{4-3k}} \right) h$$
 (A9)

When the grain is polished parallel to the c-axis and less than halfway ( $g < (W_2 + g)/2$ ), V and  $SA_\alpha$  are calculated using Eq. (A10) and (A11) which combine estimated V and surface area of the whole original grain and the removed portion of the grain approximated as half a symmetric prism broken parallel to the c-axis.

$$V = \pi h \left( ab - \frac{1}{2} a_p b_p \right) \tag{A10}$$

$$SA_{\alpha} = 2\pi \left( ab - \frac{1}{2}a_{P}b_{P} \right) + \pi h \left[ (a+b) \left( 1 + \frac{3k}{10 + \sqrt{4-3k}} \right) - \frac{1}{2}(a_{P} + b_{P}) \left( 1 + \frac{3k_{P}}{10 + \sqrt{4-3k_{P}}} \right) \right]$$
(A11)"

RC1: Line 86: Specify what the ellipsoid coefficient is.

AC: The ellipsoid coefficient is a feature of Knud Thomsen's formula to approximate ellipsoid surface area employed in Ketcham et al., (2011) and adapted here. A citation for this has been added to the text in Line 283.

RC1: Line 114-115: Same as for the ellipsoid does apply for a cylinder, it is not possible to correctly determine r when more than half the grain is gone. The equation that you are using r = min(W1, W2) will underestimate r. Why not using equation 1 to estimate the correct radius?

AC: I agree a better approach is possible. See above for revisions to the treatment of "cylindrical" grains by treating them as prisms with ellipsoidal terminations.

RC1: Line 158-186: It is unclear to me if you calculate the Ft for the whole grain or the mounted/polished grain. The Ft value of mounted/polished grains will in most cases be higher/lower than the theoretical value of the whole grain (similar only if exactly half the grain is removed).

AC:  $F_T$  is calculated using V and  $SA_\alpha$  for the polished grain fragment only. As noted above, the manuscript has been revised to clarify this.

RC1: Line 187: Please add more details on how you did the comparison, are this read data or synthetic data and give details how the methods of Ketcham and Reiners differ from your approach.

AC: Thank you for the feedback that more details are needed. The data used in the original manuscript is a real dataset acquired for a detrital zircon sample; this is now contained in Supplementary Text. The main paper discusses comparisons using a synthetic dataset. How the synthetic dataset was generated is described in Lines 163-179:

"A synthetic dataset (Table S1) was used to evaluate the protocol and compare with existing approaches to calculating FT values. The synthetic dataset was designed to test a range of original grain geometries, total grain sizes, grinding and polishing orientations, and grinding depths greater than the maximum average zircon alpha stopping distance (> ~ 18.5 µm; Ketcham et al., 2011). Total grain size was defined by a combination of "size" corresponding to the c-axis parallel length – generally the longest grain axis corresponding to grain length measurements L1 and L2, "width ratio" between the two c-axis perpendicular grain lengths (corresponding to a and b crystallographic axes and grain width measurements W1 and W2), and "aspect ratio" between the c-axis parallel and perpendicular axes lengths. First, whole, unpolished synthetic grains were created with sizes (L1 and L2) including "Smallest" (60 μm), "Small" (100 μm), "Medium" (150 µm), or "Large" (200 µm), a range of aspect ratios where the first axis (W1) was set to 0.3-1 times the size, and a range of width ratios where the second short axis  $(W_2)$  was set to 0.5 - 1 times  $W_1$ . The range of c-axis parallel sizes was chosen to reflect sizes commonly seen in natural zircon. Aspect and width ratio ranges were chosen to reflect observed ranges of these ratios while also ensuring that grinding depth would always be greater than one alpha stopping distance. This was done to ensure no complications to interpreting FT arising from incomplete removal of the alpha-ejection rim. Grains were created in this way for all common zircon idealized geometries: ellipsoid, cylinder, and tetragons with no, one, or two terminations. "Polished grains" were then created by assigning grinding depth as a fraction of the total width or length of the grain depending on whether grains were polished parallel or perpendicular to the c-axis, respectively. The range of grinding depths includes 0 (unpolished grains) and 0.25-0.75 of the total width or length."

I have also added more description of the Ketcham et al. and Reiners et al. methods:

Line 61-63: "For grains polished parallel to the c-axis Reiners et al. (2007) provides a protocol for a limited number of cases: cylindrical and orthorhombic prisms ground and polished to a depth between one alpha-stopping distance and less than half of the original c-axis perpendicular thickness of the crystal."

Line 201-206: "Although the Ketcham et al. protocol is not designed for polished grains, it might be assumed that the difference in final  $F_T$  value obtained by applying it might be negligible due to the application of the same polynomial coefficients in both methods. Here, the methods are compared to show that systematic biases are introduced when a whole-grain protocol is applied to polished grains that can result in limited utility of the dataset. This comparison was achieved by duplicating the synthetic dataset and setting grinding depth g equal to 0 for all synthetic grains so that the code treated them as unpolished for calculating V and  $SA_{\alpha}$ , and  $F_T$ ."

Line 231-244: "The Reiners et al. (2007) protocol uses V and  $SA_{\alpha}$  of grain fragments with the  $F_T$  formulas and polynomial coefficients of Farley (2002) but it applies only to cylindrical and non-terminated tetragonal grain geometries polished less than halfway through the original width of the crystal. For grains with maximum symmetry (Fig. 3c), the synthetic  $F_T$  results of the Reiners et al. protocol are almost identical to the new protocol, with all  $F_T$  values > 0.5. In these cases, the calculation of  $SA_{\alpha}$  and V are the same between the two methods and any discrepancy is the result of differences between the polynomial coefficients used. However, systematic offsets related to grain geometry appear when comparing  $F_T$  for cylindrical grains with minimal symmetry (Fig. 3d). For cylindrical grains, the Reiners et al. protocol results in higher  $F_T$  values than the new protocol reflecting that the Reiners et al. protocol assumes grains are true cylinders with symmetry about the c-axis. This results in underestimates of  $SA_{\alpha}$  and V compared to the new protocol which treats cylinders as prisms with ellipsoidal pinacoid terminations. For tetragonal grains,  $F_T$  values calculated using the new protocol are larger than values calculated using the Reiners et al. protocol. Tetragonal  $SA_{\alpha}$  and V are calculated using the same formulas regardless of protocol, so differences arise solely from

the difference in polynomial coefficients. Although there is not a significant difference in the number of grain fragments with  $F_T > 0.5$  between the new and Reiners protocols, the addition of ellipsoid grains and the greater range of grinding depths covered under the new protocol makes it an improvement over the existing Reiners et. al. method."

RC1: Line 189: Please clarify what you mean with 'uncorrected method'?

AC: The use of "uncorrected method' in this context was meant to differentiate the method of Ketcham et al. (2011) for calculating volume, surface area, and FT for whole grains from my protocol for calculating these values for polished grain fragments. As stated throughout the text, my protocol draws heavily on the approach and equations employed by Ketcham et al. but modifies them to reflect the volume and alpha-ejection surface area of polished grain fragments. However, based on this and other comments, this wording has only created confusion. The revised manuscript refers simply to the "Ketcham et al. protocol" in contrast to the "new protocol" presented in the manuscript.

RC2: First, I would suggest that the author reconsider the use of "volume and derived value" in the title, which could be more informative if it specified, for example, "improved calculation of volume, FT correction, and other derived values for polished zircon"... At the end of the day, the FT correction is what most readers are interested in.

AC: I agree, thank you for the suggestion. The title has been revised to:

"Technical Note: Improved calculation of volume, FT correction, and other derived data for polished zircon (U-Th)/Thermochronology"

RC2: Second, the manuscript would be much improved with greater discussion of the specifics of how the proposed calculations differ from previous protocols (e.g. in Lines 210-215). The equations are presented, but not much is done to demonstrate their superiority, besides Figure 2. Which cases lead to the large (95%) variation? It seems to be just a single grain or two? In what cases/geometries/grain sizes generally have minimal difference between the different protocols? Is it small grains that are particularly effected? or when very little of the grain is polished, or a lot of it is polished away?

AC: I agree that further discussion of the differences between methods would strengthen the manuscript. Notably, the conditions under which the different protocols are designed to apply are very different. I agree that this is an important point that was missing from the manuscript, and it has now been added.

Line 61-63: "For grains polished parallel to the c-axis Reiners et al. (2007) provides a protocol for a limited number of cases: cylindrical and orthorhombic prisms ground and polished to a depth between one alpha-stopping distance and less than half of the original c-axis perpendicular thickness of the crystal."

Line 201-206: "Although the Ketcham et al. protocol is not designed for polished grains, it might be assumed that the difference in final  $F_T$  value obtained by applying it might be negligible due to the application of the same polynomial coefficients in both methods. Here, the methods are compared to show that systematic biases are introduced when a whole-grain protocol is applied to polished grains that can result in limited utility of the dataset. This comparison was achieved by duplicating the synthetic dataset and setting grinding depth g equal to 0 for all synthetic grains so that the code treated them as unpolished for calculating V and  $SA_{\alpha}$ , and  $F_T$ ."

Line 231-244: "The Reiners et al. (2007) protocol uses V and  $SA_{\alpha}$  of grain fragments with the  $F_T$  formulas and polynomial coefficients of Farley (2002) but it applies only to cylindrical and non-terminated tetragonal grain geometries polished less than halfway through the original width of the crystal. For grains with maximum symmetry (Fig. 3c), the synthetic  $F_T$  results of the Reiners et al. protocol are almost identical to the new protocol, with all  $F_T$  values > 0.5. In these cases, the calculation of  $SA_{\alpha}$  and V are the same between the two methods and any discrepancy is the result of differences between the polynomial coefficients used. However, systematic offsets related to grain geometry appear when comparing  $F_T$  for cylindrical grains with minimal symmetry (Fig. 3d). For cylindrical grains, the Reiners et al. protocol results in higher  $F_T$  values than the new protocol reflecting that the Reiners et al. protocol assumes grains are true cylinders with symmetry about the c-axis. This results in underestimates of  $SA_{\alpha}$  and V compared to the new protocol which treats cylinders as prisms with ellipsoidal pinacoid terminations. For tetragonal

grains,  $F_T$  values calculated using the new protocol are larger than values calculated using the Reiners et al. protocol. Tetragonal  $SA_{\alpha}$  and V are calculated using the same formulas regardless of protocol, so differences arise solely from the difference in polynomial coefficients. Although there is not a significant difference in the number of grain fragments with  $F_T > 0.5$  between the new and Reiners protocols, the addition of ellipsoid grains and the greater range of grinding depths covered under the new protocol makes it an improvement over the existing Reiners et. al. method."

Regarding why the protocols differ, I have addressed this by adding protocol evaluation using a synthetic dataset that includes many more possible scenarios than the real dataset originally presented. This is now in the main text of the manuscript, while the real data example has been moved to a supplement. Grain size and grain symmetry are the dominant factors in how different the results of the different protocols are to each other. Large, symmetric grains give relatively similar results regardless of protocol, while smaller, less symmetric grains differ more significantly. This is shown and discussed in a completely new Section 3 and two figures (Lines 162-258; Fig. 2, 3).

RC2: A histogram could be provided showing the % difference between the different protocols for each grain in the dataset. Is there a systematic skew towards overestimating or underestimating FT?

AC: I agree showing any apparent trends in the difference between protocols would be useful. Rather than adding histograms, I have opted to for minimal figures but I have added discussion of when and why protocols are the same and different and annotations to Figure 3 to show how trends in the difference between protocols with increased grinding depth.

Line 206-215: "For the most symmetric grains (aspect and width ratios of 1, Fig. 3a), applying the whole-grain Ketcham et al. protocol results in  $F_T$  values that are generally lower than the new protocol. This is expected: the Ketcham et al. protocol calculates  $SA_{\alpha}$  that is higher than the real polished  $SA_{\alpha}$  in all cases, and in the case of ellipsoids calculates V that is significantly smaller than the real polished V. If the recommended 0.5  $F_T$  cutoff for accepting (U-Th)/He analyses is applied (e.g., Flowers et al., 2022b) to the polished fragments, use of the Ketcham et al. protocol would result in rejection of more ellipsoid, cylindrical, and terminated tetragonal grains than the new protocol while more non-terminated tetragonal grains would be kept. This is because the only difference between the two protocols for non-terminated tetragonal grains is the inclusion of the polished face in  $SA_{\alpha}$ . For grains with the least symmetry, both protocols result in in the rejection of most grain fragments, but the Ketcham et al. protocol results in more total rejections due to its inaccurate estimates of V and  $SA_{\alpha}$ ."

Line 233-243: "For grains with maximum symmetry (Fig. 3c), the synthetic  $F_T$  results of the Reiners et al. protocol are almost identical to the new protocol, with all  $F_T$  values > 0.5. In these cases, the calculation of  $SA_\alpha$  and V are the same between the two methods and any discrepancy is the result of differences between the polynomial coefficients used. However, systematic offsets related to grain geometry appear when comparing  $F_T$  for cylindrical grains with minimal symmetry (Fig. 3d). For cylindrical grains, the Reiners et al. protocol results in higher  $F_T$  values than the new protocol reflecting that the Reiners et al. protocol assumes grains are true cylinders with symmetry about the c-axis. This results in underestimates of  $SA_\alpha$  and V compared to the new protocol which treats cylinders as prisms with ellipsoidal pinacoid terminations. For tetragonal grains,  $F_T$  values calculated using the new protocol are larger than values calculated using the Reiners et al. protocol. Tetragonal  $SA_\alpha$  and V are calculated using the same formulas regardless of protocol, so differences arise solely from the difference in polynomial coefficients."

RC2: Notably, Fig. 2 only shows that these protocol are different. But it's not immediately clear why, practically speaking, the additional complication of assigning grains to particular sub- classes of geometries and degrees of polishing based on limited 2D measurements from a polished mount wouldn't simply be introducing more assumptions and/or errors. I could certainly imagine how these detailed calculations here could be better - but I don't think that's necessarily the case, and the author needs to demonstrate that.

For example, the author states that "The Reiners et al. method of accounting for FT corrections due to polishing, while frequently resulting in < 5 % difference, can also vary more significantly, likely due to simplifying assumptions made by this method regarding grain geometry, orientation, and depth of polishing." The author should expand on this sentence and explicitly discuss those simplifying assumptions. How and when exactly do they vary so significantly? And most importantly, for many readers, the question is whether it is practical to move beyond those simplifying

assumptions. It would be helpful if the author distinguished the specifics of the cases (e.g. the one with the 92% difference) that led to the large difference. Why and how are the approximations that are used here (particularly for the c-axis parallel cases) better/different than the simplifying assumptions used by others?

AC: I agree that further discussion and justification of this method compared to others was necessary. I have addressed this with the use of the synthetic dataset to show the dominant factors that control differences between methods and how more accurately calculating V and  $SA_{\alpha}$  through the added complication of geometry classification and measurements results in more usable grains and thus more data. I have also revised the text in the Introduction, Section 2, and Section 3 to be explicit about the fact that existing methods do not apply to many common scenarios.

Line 59-72: "Although this approach can be applied to any crystal geometry and plane of symmetry, the He and Reiners protocol is focused on cylindrical grains polished or broken perpendicular to the c-axis. For grains polished parallel to the c-axis Reiners et al. (2007) provides a protocol for a limited number of cases: cylindrical and orthorhombic prisms ground and polished to a depth between one alpha-stopping distance and less than half of the original c-axis perpendicular thickness of the crystal. In reality, zircon encompass a range of morphologies depending on lithology and geologic history which can be approximated as cylinders, ellipsoids, and orthorhombic prisms with or without pyramidal terminations (commonly referred to as "tetragonal" geometries even when a- and b-axis measurements are not equivalent). The grinding and polishing orientation of individual crystals can be parallel or perpendicular to the crystallographic c-axis, and because of the natural variation in crystal size, it is common for polishing to remove a variable amount of crystal when multiple crystals are mounted and prepared together (e.g., Fig la). Protocols to determine FT corrections and other derived data for polished zircon based on geometry and volume must therefore encompass these different scenarios in order to maximize the number of grains that can be used for analysis in a given sample and grain mount. To address the lack of a comprehensive approach to volume-derived data for polished zircon, this contribution presents a protocol..."

Line 201-217: "Although the Ketcham et al. protocol is not designed for polished grains, it might be assumed that the difference in final FT value obtained by applying it might be negligible due to the application of the same polynomial coefficients in both methods. Here, the methods are compared to show that systematic biases are introduced when a whole-grain protocol is applied to polished grains that can result in limited utility of the dataset. This comparison was achieved by duplicating the synthetic dataset and setting grinding depth g equal to 0 for all synthetic grains so that the code treated them as unpolished for calculating V and SA\alpha, and FT. For the most symmetric grains (aspect and width ratios of 1, Fig. 3a), applying the whole-grain Ketcham et al. protocol results in FT values that are generally lower than the new protocol. This is expected: the Ketcham et al. protocol calculates  $SA_{\alpha}$  that is higher than the real polished  $SA_{\alpha}$ in all cases, and in the case of ellipsoids calculates V that is significantly smaller than the real polished V. If the recommended 0.5 FT cutoff for accepting (U-Th)/He analyses is applied (e.g., Flowers et al., 2022b) to the polished fragments, use of the Ketcham et al. protocol would result in rejection of more ellipsoid, cylindrical, and terminated tetragonal grains than the new protocol while more non-terminated tetragonal grains would be kept. This is because the only difference between the two protocols for non-terminated tetragonal grains is the inclusion of the polished face in  $SA_{\alpha}$ . For grains with the least symmetry, both protocols result in the rejection of most grain fragments, but the Ketcham et al. protocol results in more total rejections due to its inaccurate estimates of V and  $SA_{\alpha}$ . This is important for real datasets in which grain aspect and width ratios can be expected to vary widely and rarely match the maximum symmetry case. By taking grinding and polishing into account, the new protocol results in FT values that reflect the true  $SA_{\alpha}$  and V of the measured grain fragment and are more likely to meet the criteria for being accepted."

Line 231-244: "The Reiners et al. (2007) protocol uses V and  $SA_{\alpha}$  of grain fragments with the  $F_T$  formulas and polynomial coefficients of Farley (2002) but it applies only to cylindrical and non-terminated tetragonal grain geometries polished less than halfway through the original width of the crystal. For grains with maximum symmetry (Fig. 3c), the synthetic  $F_T$  results of the Reiners et al. protocol are almost identical to the new protocol, with all  $F_T$  values > 0.5. In these cases, the calculation of  $SA_{\alpha}$  and V are the same between the two methods and any discrepancy is the result of differences between the polynomial coefficients used. However, systematic offsets related to grain geometry appear when comparing  $F_T$  for cylindrical grains with minimal symmetry (Fig. 3d). For cylindrical grains, the Reiners et al. protocol results in higher  $F_T$  values than the new protocol reflecting that the Reiners et al. protocol assumes grains are true cylinders with symmetry about the c-axis. This results in underestimates of  $SA_{\alpha}$  and V compared to the new protocol which treats cylinders as prisms with ellipsoidal pinacoid terminations. For tetragonal grains,  $F_T$  values calculated using the new protocol are larger than values calculated using the Reiners et al. protocol.

Tetragonal  $SA_{\alpha}$  and V are calculated using the same formulas regardless of protocol, so differences arise solely from the difference in polynomial coefficients. Although there is not a significant difference in the number of grain fragments with  $F_T > 0.5$  between the new and Reiners protocols, the addition of ellipsoid grains and the greater range of grinding depths covered under the new protocol makes it an improvement over the existing Reiners et. al. method."

RC2: When polished perpendicular to the c-axis, the calculations would essentially be the same case as the fragmentation correction for grains with one end broken, which we discussed in a similar paper (He and Reiners, 2022). For these cases, I imagine the modified FT would be exactly the same as the protocol propose here?

AC: Yes, the approach is the same in which the polished fragment has an FT correction equivalent to a whole crystal with length 2x the fragment length and I apologize for not citing this He and Reiners (2022) in regards to this approach. The Introduction and Section 2 have been revised to include description of the He and Reiners (2022) approach and shared aspects with the new protocol.

Line 56-61: "... the same symmetry logic can be applied to crystals broken perpendicular to an axis of symmetry when the true original axis length is unknown (He and Reiners, 2022). The broken interior face of the crystal is treated as a plane of symmetry such that the fragment has the same  $F_T$  as a whole grain with an axis length double the axis length of the fragment. V and  $SA_{\alpha}$  of the fragment can be calculated by dividing the V and  $SA_{\alpha}$  of the reconstructed whole grain in half. Although this approach can be applied to any crystal geometry and plane of symmetry, the He and Reiners protocol is focused on cylindrical grains polished or broken perpendicular to the c-axis."

Line 92-96: "In most cases, calculating these values is accomplished by adopting the same approach as He and Reiners (2022) in which the polished grain is treated as a crystal broken along a plane of symmetry such that V and  $SA_{\alpha}$  of the polished fragment are half of a whole "assumed grain" created by reflecting the existing fragment across the plane of polishing (Fig. 1c).  $F_T$  of the fragment is thus equal to  $F_T$  of the assumed grain."

RC2: Finally, something additional that would be relevant to add in the discussion: the idea that the SA/V of polished grains can be used to modify FT corrections assumes that the polynomial function relating SA/V to FT is nearly identical for most geometries. But it is not entirely identical, and polished grains would deviate pretty far from ideal geometries used to determine the SA/V-FT functions.

AC: I agree that this is worth discussing. I based my protocol on  $F_T$  equations from Ketcham et al. (2011) rather than Farley (2002) precisely because the Ketcham et al. equations are tailored to specific geometries. I have added a paragraph to Section 2 discussing in general how  $F_T$  is calculated and polynomial coefficients.

Line 123 - 140: " $F_T$  depends not just on volume, but also on  $SA_{\alpha}$ , dependence which is represented here using the term  $R_{SV}$ , or volume-to-surface-area equivalent spherical radius, calculated using Eq. (2) as in Ketcham et al. (2011).

$$R_{SV} = \frac{3V}{SA_{\alpha}} \tag{2}$$

 $R_{SV}$  serves the same function as the  $\beta$  term introduced by Farley (2002) to relate grain measurements to  $F_T$  via a polynomial function with the general form of Eq. (3).

$$F_T = 1 + a_1 \beta + a_2 \beta^2 + a_3 \beta^3 + \dots$$
 (3)

Polynomial coefficients a1, a2, and a3, etc. are determined via series of Monte Carlo simulations of variable grains and fitting the results (e.g., Hourigan et al., 2005; Ketcham et al., 2011) and depend on alpha-stopping distance and grain geometry. For the new protocol, the FT equations and coefficients of Ketcham et al. (2011) are adopted as the basis for calculating FT (Appendix A) because they are fit to the full range of grain geometries commonly seen in natural zircon. For grains that begin whole as ellipsoids, cylinders, or tetragons without terminations, grinding and polishing results in remaining grain fragments with morphologies that are still well-described by the original geometries and FT equations tailored to those geometries, such that minimal uncertainty is introduced by applying the geometry-specific coefficients of Ketcham et al. to these polished grains. The whole-grain coefficients are likely less applicable to grain geometries that change more significantly with grinding and polishing, namely tetragonal geometries with one or two terminations, and the new protocol should be applied with caution to these geometries. However, even with this

limitation, the new protocol improves on existing protocols for polished grain FT values through the addition of ellipsoid geometries and a range of polishing depth beyond half of the original grain width."

RC2: It's not clear from Fig. 1 what the different labels (e.g. w1 wp) are referring to in many of the diagrams.

AC: Thank you for the feedback that this is unclear.  $W_1$ ,  $W_2$ ,  $L_1$ , and  $L_2$  are the measurements made of the grain while a, b, c, and h refer to the axes, semi-axes, and height of the idealized geometry. The figure is meant to show visually how these relate to each other to compliment the mathematical relationships between measurements and geometric parameters given in Table 1. Sets of  $W_1$ ,  $L_1$  and  $W_2$ ,  $L_2$  measurements are made by rotating the grains 90° such that these measurements can also define a rectangular prism that surrounds the grain, shown in revised Figure 1. To help clarify these relationships I have added text to Section 2

Line 83-85: "Like standard approaches for calculating whole-grain V (e.g., Zeigler et al., 2024), two orthogonal sets of length and width measurements ( $L_1$ ,  $W_1$  and  $L_2$ ,  $W_2$ ), are made by rotating the grain fragment (Fig. 1b)."

RC2: What you call SA is not actually surface area - but rather something like the alpha-ejection- affected-surface area. I suggest using a subscript to clarify this (SA $\alpha$ ), or something similar, as it is can be confusing to readers. Note that in He and Reiners 2022, we called  $\beta\alpha$  =the ratio of alpha-ejection-affected surface area to volume.

AC: This is true, and this modification of "SA" from the true SA is explained in Line 89-90. To further distinguish alpha-ejection surface area from total surface area I have adopted your suggestion of using an alpha subscript throughout the text. I have also added explanation of how  $R_{\rm SV}$  relates to  $\beta$  and the relationship of both to  $F_{\rm T}$ .

Line 123 - 129: " $F_T$  depends not just on volume, but also on  $SA_{\alpha}$ , dependence which is represented here using the term  $R_{SV}$ , or volume-to-surface-area equivalent spherical radius, calculated using Eq. (2) as in Ketcham et al. (2011).

$$R_{SV} = \frac{3V}{SA_{\alpha}} \tag{2}$$

 $R_{SV}$  serves the same function as the  $\beta$  term introduced by Farley (2002) to relate grain measurements to  $F_T$  via a polynomial function with the general form of Eq. (3).

$$F_T = 1 + a_1 \beta + a_2 \beta^2 + a_3 \beta^3 + \dots {3}$$

RC2: There should be more details about the test dataset: what was the measurement protocol? the range of grain sizes? how were the grains assigned into different geometries if they were already polished?

AC: The explanation of how grains are measured and classified by geometry has been expanded in Section 2.

Lines 79-87: "The protocol presented here adapts existing approaches for determining whole-grain V,  $SA_{\alpha}$ , and  $F_T$  for ground and polished grain fragments. First, the polished grains are removed from the mounting medium and inspected and measured using a binocular microscope with digital camera and microscope imaging software. Grains are classified as ellipsoidal, cylindrical, or "tetragonal" geometries, which can include two, one, or no pyramidal terminations. In order to be classified as cylindrical or tetragonal, the unpolished part of the grain must include visible crystal faces that are unrelated to the polished face. For cylinders, these faces are only perpendicular to the long axis while for tetragonal grains, some must be parallel to the long axis (Fig 1). If there are no observed crystal faces, the grain is classified as an ellipsoid. Like standard approaches for calculating whole-grain V (e.g., Zeigler et al., 2024), two orthogonal sets of length and width measurements ( $L_1$ ,  $W_1$  and  $L_2$ ,  $W_2$ ), are made by rotating the grain fragment (Fig. 1b)."

The description and discussion of the real dataset has been moved to a Supplement, and I have updated it with inclusion of grain size distribution and other descriptors.

Lines S21-S31: "The real dataset (n = 70, Table S2) includes a much more limited sampling of original grain geometries, grain sizes, aspect ratios, and width ratios than the synthetic dataset. Geometries include ellipsoids (n = 70, Table S2) includes a much more limited sampling of original grain grain grain grain sizes, aspect ratios, and width ratios than the synthetic dataset. Geometries include ellipsoids (n = 70, Table S2) includes a much more limited sampling of original grain gra

58), cylinders (n = 8), non-terminated tetragons (n = 3) and tetragons with one termination (n = 1). The high number of ellipsoidal grains is unsurprising given grain abrasion that occurs during sediment transport. Size (c-axis parallel length) ranges from 102 to 361.1 um with a median size of 199.3. Aspect ratio (the ratio between the c-axis parallel length and larger perpendicular axis) ranges from 0.3209 to 2.5222 with a median of 1.0669. Width ratio (the ratio between the two c- axis perpendicular axes) ranges from 0.3303 to 0.9929 with a median of 0.6528. The aspect ratios and width ratios present in the real dataset are similar to the ranges tested with the synthetic dataset. However, the real dataset only includes zircon with c-axis parallel size in the higher end of the range, or higher than sizes tested with the synthetic dataset. Grinding depth was significantly, calculated using measurements of mounted glass beads as in Section 2 of the main text, was significantly higher than the zircon average alpha stopping distance in all cases (Table S2)."

All grain measurements  $L_1$ ,  $L_2$ ,  $W_1$ ,  $W_2$ , and  $R_{SV}$  and  $R_{FT}$  proxies for grain size for each individual grain are included in Table S2.

---

## Referee Report (RR1)

Reviewer comments.

Having read the manuscript and the two previous reviewers' comments I feel that the author made the necessary changes requested by the reviewers and editor. I believe that the clarity of the manuscript has been much improved. This improvement was required to ensure readers understood the experimental process suggested by the author which could be followed using the program and equations within.

However, I do have some minor corrections that if applied I believe would improve the manuscript. I leave whether they are done or not to the discretion of the editor and author.

Line 9: obtain the maximum geologically relevant information.

Could this be clarified, does this sentence mean the most relevant information or the largest amount of relevant information?

Line 110: This calculation requires additional measurements of the polished grain surface: length  $(L_P)$  and width  $(W_P)$  of the polished face.

If the grinding depth is assumed to be the amount of crystal removed and the crystal is assumed to have been polished parallel to the c-axis then the grain surface does not need to be measured and can be calculated from the assumed geometry.

This can be done for both the ellipsoid and the tetragonal prism.

I think this could lead to improvements to the manuscript in three ways.

Firstly, as the author themselves appears to state, it is often difficult to differentiate faces from crystal exteriors using light microscopy. This would reduce the burden on the experimentalist and speed up the process of measuring crystal sizes which is a significant time sink.

Secondly, because it is often indistinguishable, then calculating these values from a grinding depth which is used anyway, would increase the accuracy of the resultant model.

Unless, this measurement is in fact being used to account for the fact that the model geometries might not match with the actual geometries in the crystal in reality. In which case this should be stated.

Thirdly it might mean that in Figure 1, the measurements of Lp and Wp could be removed from the figure, and make it easier to read.

Line 206: For the most symmetric grains...

Here I would suggest replacing symmetric with equidimensional or equant. As crystals could be equant but not symmetric, such as would be the case with tetragonal prisms with a single termination. This would lead to changes in lines 213 and figure captions 2 and 3.

Figure 2: Do the the 'least symmetric' grains which pass the requirement of Ft > 0.5 have quite variable shapes? I think it would be nice if those shapes were added as a 3D mini drawing to the graphs. This might help contextualise the 'least symmetric' data.

Figure 3: In the caption it is written that a) is for equant crystals and that section b) is for minimally equant crystals – could it also be put on the figure? Again, maybe as shape drawings?

I also think that there are several assumptions that underlie the method that are not explicitly stated, or perhaps I have missed them. This list is not exhaustive but some might be:

- a) That the spheres used for measuring grinding depth and the zircon crystals are at the same depth under the epoxy surface
- b) That the zircon crystal is not at an angle to the polishing surface
- c) That the crystals have homogenous parent isotope concentrations
- d) That the polynomial coefficients used to calculate Ft from SA/V are used while perhaps actually deviating from the geometries which they were defined from (this assumption was highlighted by reviewer 2 and the author addresses it in Lines 137-140)

My personal preference is for assumptions like these, whether they are likely to cause inaccuracies or not, to be focussed in one area of the method section.

---

## Referee Report (RR2)

Technical Note: Improved calculation of volume, FT correction, and other derived data for polished zircon (U-Th)/He thermochronology

Barra Peak

Gchron-2024-33

**Referee Report**

This manuscript highlights the importance of accurate Ft and aliquot volume calculations in (U-Th)/He Thermochronology. The author provides improved equations to calculate Ft and grain volume accounting for grain polishing required to make in situ thermochronologic measurements. The author greatly improved the manuscript between the first and second round of revisions. I greatly appreciate the author's addition of a comprehensive synthetic dataset to demonstrate the effectiveness of the proposed protocol compared to former methods. I believe this work represents a substantial contribution to our field, and should be published following minor revisions to address readability.

Overall, the subject matter and scientific process in this manuscript are well presented. The manuscript could benefit from some minor edits to improve the flow. The introduction section, largely changed from the original submission, is somewhat disorganized. Much of the information presented in this section focuses on the technicalities of other methods for measuring V and Ft, which may be better discussed in section 2.

**Line 9:** "...maximum geologically relevant information" wording is clumsy and missing a preposition?

Line 17: You do not specify the acronym eU as effective uranium in your definition (Line 12)

**Line 75-77:** It is unclear what the subject of this sentence is. What is being directly compared to Ft corrections of whole grains?

**Lines 74-104:** This paragraph covers a lot of information (some of which might do better in the discussion, i.e. lines 87-89), and could be broken down to streamline the story. I recommend having a paragraph that outlines previous work and limitations followed by a paragraph describing your proposed methodology and how it addresses these limitations.

**Table 1:** Caption should include mention of where on the grain 2D measurements are taken on the grain (the polished surface?).

**Line 524:** Have you tried incorporating measurements of tetragonal grain "tippiness" or the pyramidal length measured parallel to c-axis from tip to base of the pyramid? The aspect ratio of these pyramidal terminations can greatly affect grain volume and surface area calculations.

**Line 587:** Section numbering is incorrect

---

## Author Response (AR2)

**GChron-2024-33:** Technical Note: Improved calculation of volume, FT correction, and other derived data for polished zircon (U-Th)/He thermochronology

**Author's Response to Reviews, Iteration: Minor Revision**Barra Peak

AC: Firstly, I would like to thank the reviewers for agreeing to review a manuscript that had already been heavily revised since initial submission and taking the time to review the first-round manuscript and reviewer comments in addition to providing new feedback on the revised version. I appreciate their feedback on where the manuscript still needs clarification and have adopted the majority of their suggestions. Responses to specific feedback are below.

**RC3** Reviewer comments:**

**RC3**: Having read the manuscript and the two previous reviewers' comments I feel that the author made the necessary changes requested by the reviewers and editor. I believe that the clarity of the manuscript has been much improved. This improvement was required to ensure readers understood the experimental process suggested by the author which could be followed using the program and equations within.

However, I do have some minor corrections that if applied I believe would improve the manuscript. I leave whether they are done or not to the discretion of the editor and author.

**RC3**: Line 9: *obtain the maximum geologically relevant information*. Could this be clarified, does this sentence mean the most relevant information or the largest amount of relevant information?

**AC**: The sentence was meant to refer to the latter reading. Based on this comment and a comment from RC4, I have modified the sentence to be more explicit (Lines 8-10).

"Polishing mounted zircon crystals prior to bulk grain (U-Th)/He thermochronology analysis provides opportunities for characterizing and subsampling each grain via in situ methods to obtain additional information relevant for (U-Th)/He date interpretation and the broader geologic questions of interest."

**RC3**: Line 110: This calculation requires additional measurements of the polished grain surface: length (LP) and width (WP) of the polished face.

If the grinding depth is assumed to be the amount of crystal removed and the crystal is assumed to have been polished parallel to the c-axis then the grain surface does not need to be measured and can be calculated from the assumed geometry. This can be done for both the ellipsoid and the tetragonal prism. I think this could lead to improvements to the manuscript in three ways.

**AC**: The reviewer is correct that this can be done for tetragons, and this is in fact done and reflected in Table 1, Appendix A, and the accompanying R script. The notations on Figure 1 were mistakenly left from an earlier version of the manuscript and this has now been corrected. Cylinders and ellipsoids need the additional measurements of the polished face to account for the curvature of the exterior surface which is not calculable from the four orthogonal measurements L1, L2, W1, and W2 alone due to the fact that  $a \ne b$ . I have clarified this in Line 125:

"This calculation requires additional measurements of the polished grain surface for ellipsoid and cylinder geometries..."

**RC3**: Firstly, as the author themselves appears to state, it is often difficult to differentiate faces from crystal exteriors using light microscopy. This would reduce the burden on the experimentalist and speed up the process of measuring crystal sizes which is a significant time sink.

**AC**: I'm not entirely sure what the reviewer is referring to with this comment. In Line 129 I state that  $L_P$  and  $W_P$  are often indistinguishable from  $L_1$  and  $W_1$  but this is unrelated to crystal faces. This is just referring to the fact that these measurements are often similar/overlap. It is easy to imagine situations when this would not be the case though, such as when a very large grain is only abraded a small amount. I have edited Line 131 to try to clarify this:

"In practice, LP and WP are often indistinguishable from L1 and W1 for small and medium grains, but for larger grains, the difference between the polished face and total axis measurements can be much greater."

In regards to time, the experimentalist is already making 4 measurements per grain so adding 1-2 more for a subset of grains does not really add that much time to the total endeavor.

**RC3**: Secondly, because it is often indistinguishable, then calculating these values from a grinding depth which is used anyway, would increase the accuracy of the resultant model. Unless this measurement is in fact being used to account for the fact that the model geometries might not match with the actual geometries in the crystal in reality. In which case this should be stated.

**AC**: As noted above, measurements of the face are needed to calculate the values  $a_P$ ,  $b_P$ , and  $c_P$  in cases when  $L_P$  and  $W_P$  are not indistinguishable from  $L_1$  and  $W_1$ . The only way to determine if they are indistinguishable is to measure the face, so personally I think it is better to have these measurements be a standard part of the protocol.

**RC3**: Thirdly it might mean that in Figure 1, the measurements of Lp and Wp could be removed from the figure, and make it easier to read.

**AC**: The LP and WP measurements on the tetragon subfigure in Figure 1b were included by mistake and have been removed. As explained above, these measurements are needed for ellipsoids and cylinders and are retained.

**RC3**: Line 206: For the most symmetric grains...

Here I would suggest replacing symmetric with equidimensional or equant. As crystals could be equant but not symmetric, such as would be the case with tetragonal prisms with a single termination. This would lead to changes in lines 213 and figure captions 2 and 3.

**AC**: This suggestion has been adopted throughout Section 4 (Lines 244, 251, 254, 278, 281, 283 and Figure 2 and 3 captions).

**RC3**: Figure 2: Do the 'least symmetric' grains which pass the requirement of Ft > 0.5 have quite variable shapes? I think it would be nice if those shapes were added as a 3D mini drawing to the graphs. This might help contextualise the 'least symmetric' data.

**AC**: The least symmetric grains do vary in terms of relationship between the a, b, and c axes. I have added a cartoon depicting this as Figure 2b.

**RC3**: Figure 3: In the caption it is written that a) is for equant crystals and that section b) is for minimally equant crystals – could it also be put on the figure? Again, maybe as shape drawings?

**AC**: I have referenced the new Figure 2b regarding the variability in shapes in the Figure 3 caption.

**RC3**: I also think that there are several assumptions that underlie the method that are not explicitly stated, or perhaps I have missed them. This list is not exhaustive but some might be:

a) That the spheres used for measuring grinding depth and the zircon crystals are at the same depth under the epoxy surface

**AC**: Yes, this is assumed and is generally a good assumption based on how these mounts are made. The grains are also assumed to be at the same level (touching) the epoxy surface. I have modified Figure 1a to include a glass bead for reference that shows this orientation.

**RC3**: b) That the zircon crystal is not at an angle to the polishing surface

**AC**: Yes, this is also assumed but I think this is communicated by the fact that the protocol is explicitly designed for grains polished parallel or perpendicular to the crystal c axis as shown in Figure 1 and discussed in Section 3. I have added a sentence to be explicit about the classification of polishing orientation, Line 106:

"Polishing orientation is also classified as perpendicular or parallel to the crystallographic c-axis based on visual inspection of the grain fragment."

**RC3:** c) That the crystals have homogenous parent isotope concentrations

**AC:** Yes, this is an assumption, which is made commonly for single-aliquot (U-Th)/He. Deviation from this assumption could be problematic but no more so than for whole grains. I have added a sentence being explicit about this to Section 3 (Line 144):

"Like most whole-aliquot (U-Th)/He thermochronology data reduction, the grains are assumed to have homogenous parent isotope concentrations (e.g., no zonation). Deviation from this assumption would impact the calculated dates in similar ways to zonation in whole grains (e.g., Danišík et al., 2017; Hourigan et al., 2005)."

**RC3**: d) That the polynomial coefficients used to calculate Ft from SA/V are used while perhaps actually deviating from the geometries which they were defined from (this assumption was highlighted by reviewer 2 and the author addresses it in Lines 137-140)

**AC**: As the reviewer notes, this assumption has already been addressed.

**RC3**: My personal preference is for assumptions like these, whether they are likely to cause inaccuracies or not, to be focused in one area of the method section.

**AC**: These assumptions are addressed in Section 3.

**RC4 Referee Report:**

**RC4**: This manuscript highlights the importance of accurate Ft and aliquot volume calculations in (U-Th)/He Thermochronology. The author provides improved equations to calculate Ft and grain volume accounting for grain polishing required to make in situ thermochronologic measurements. The author greatly improved the manuscript between the first and second round of revisions. I greatly appreciate the author's addition of a comprehensive synthetic dataset to demonstrate the effectiveness of the proposed protocol compared to former methods. I believe this work represents a substantial contribution to our field, and should be published following minor revisions to address readability.

Overall, the subject matter and scientific process in this manuscript are well presented. The manuscript could benefit from some minor edits to improve the flow. The introduction section, largely changed from the original submission, is somewhat disorganized. Much of the information presented in this section focuses on the technicalities of other methods for measuring V and Ft, which may be better discussed in section 2.

**AC**: I appreciate the reviewer's time providing text edits and feedback. While I agree that the level of technical background presented in the Introduction is high, I believe that this is appropriate for a Technical Note paper as it serves to explain why the new method presented is novel. It was also my hope to make this manuscript accessible to thermochronology users who may be less familiar with the mechanics of FT corrections and (U-Th)/He thermochronology derived data calculations more generally. Many geoscientists who use thermochronology data get their data already processed from the lab that generated it and the knowledge of how this was done may not be front of mind when assessing if the new method presented here is relevant for them. However, these thermochronology users can also benefit from the methodology presented in this manuscript.

**RC4: Line 9:** "...maximum geologically relevant information" wording is clumsy and missing a preposition?

**AC**: Based on this comment and a comment from the other reviewer, I have modified the sentence to be more explicit. (Lines 8-10)

"Polishing mounted zircon crystals prior to bulk grain (U-Th)/He thermochronology analysis provides opportunities for characterizing and subsampling each grain via in situ methods to obtain additional information relevant for (U-Th)/He date interpretation and the broader geologic questions of interest."

**RC4:** Line 17: You do not specify the acronym eU as effective uranium in your definition (Line 12)

**AC**: Thank you for pointing out this oversight, the acronym has been specified in Line 12.

**RC4:** Line 75-77: It is unclear what the subject of this sentence is. What is being directly compared to Ft corrections of whole grains?

**AC**: This sentence has been edited for clarification (Line 64):

"These contributions have largely focused on direct comparisons between FT corrections for polished grain fragments and FT corrections for corresponding whole crystals from which polished grains were derived..."

**RC4:** Lines 74-104: This paragraph covers a lot of information (some of which might do better in the discussion, i.e. lines 87-89), and could be broken down to streamline the story. I recommend having a paragraph that outlines previous work and limitations followed by a paragraph describing your proposed methodology and how it addresses these limitations.

**AC**: I agree this paragraph could be broken up for easier readability but I disagree that this information does not belong in the beginning of the paper. As noted above, since this is a technical note, I think it is necessary to establish how the new method differs from previous ones, which is difficult to do effectively without providing some background on previous methodology.

I have moved the bulk of Lines 74-104 to a new Section 2, "Existing methods and limitations for polished grains" (Lines 60-86) and revised the end of the Introduction (Lines 50-59):

"Although some previous work has addressed the effect of grinding and polishing on  $F_T$  corrections (He and Reiners, 2022; Marsden et al., 2021; Reiners et al., 2007), these contributions do not address other data derived from volume and surface area and apply to specific cases that do not reflect the full range of real zircon samples or sample preparation. To address the lack of a comprehensive approach to volume-derived data for polished zircon, this contribution presents a protocol and set of equations (Appendix A) coded in an R script (Code Availability) that can be integrated with existing workflows for grain characterization and (U-Th)/He thermochronology data reduction and interpretation. Values calculated under this protocol include V,  $SA_{\alpha}$ , volume-to-alpha-ejection-surface-area-equivalent spherical radius ( $R_{SV}$ ), mass (M), parent isotope concentrations, eU,  $F_T$ , and  $F_T$ -equivalent spherical radius ( $R_{FT}$ ). Results of using the protocol are evaluated using a synthetic dataset encompassing a range of possible grain geometries, sizes, polishing orientations and grinding depths (Section 4) and application to a real detrital zircon dataset (Supplementary Text, Table S2)."

**RC4:** Table 1: Caption should include mention of where on the grain 2D measurements are taken on the grain (the polished surface?).

**AC**: Most measurements are made parallel to an orthogonal set of crystal axes (e.g., c and a). This is shown in Figure 1b and has been clarified in Line 105:

"two orthogonal sets of length and width measurements ( $L_1$ ,  $W_1$  and  $L_2$ ,  $W_2$ ) parallel to orthogonal crystal axes, are made by rotating the grain fragment (Fig. 1b)."

Measurements  $L_P$  and  $W_P$  are made on the polished surface, which is explained in Line 132 and shown in Fig. 1b.

The caption for Table 2 has been updated to reference Fig. 1b.

**RC4:** Line **524:** Have you tried incorporating measurements of tetragonal grain "tippiness" or the pyramidal length measured parallel to c-axis from tip to base of the pyramid? The aspect ratio of these pyramidal terminations can greatly affect grain volume and surface area calculations.

**AC**: I agree that "tippiness" can greatly influence these values, however, I have not incorporated this as the FT polynomial coefficients assume symmetric terminations that are slope 45° to the a-b crystal plane (Ketcham et al., 2011). This simplification likely explains some of the difficulty in fitting FT to real values for tetragonal grains, both for whole grains, as documented by Ketcham et al., and for the polished grains presented here. I do not think this manuscript can really provide a better way for dealing with this limitation, but I agree it should be discussed. I have added it to discussion of results (Lines 233-236):

"Terminations are approximated using a uniform assumption of symmetric pyramidal terminations sloped 45° to the prismatic core of the grain (Ketcham et al., 2011) which is also likely responsible for some of the unexpected behavior of these grains, as in reality this angle can vary from zircon to zircon."

RC4: Line 587: Section numbering is incorrect

**AC**: This has been corrected to Section 5.

**Additional Changes:**

**Remarks from the preceding review file validation:**

Please number the sections of the supplement according to the guidelines (see "Supplements" at https://www.geochronology.net/submission.html#assets), i.e. "S1" instead of "1".

AC: This change has been made in the supplemental text file and in references to the supplement in the main text.